# Stochastic poromechanical analysis forecasts a notable exceedance probability for the 2017 Pohang, South Korea, $M_\text{w}$ 5.5 earthquake

Haiqing Wu [1,2,3,6] ✉, Victor Vilarrasa [3], Francesco Parisio[3,4], Andrés Alcolea [5], Peter Meier[5], Jesus Carrera[2,4] & Maarten Saaltink[1,2]

The 2017 Pohang $M_\text{w}$ 5.5 earthquake is currently the largest seismic event induced by Enhanced Geothermal Systems. The high uncertainty on geological and mechanical conditions of the rupture fault of this earthquake has originated a debate on its triggering mechanisms. Here, we propose a stochastic poromechanical analysis approach that combines Monte Carlo sampling and poromechanical models to address the uncertainty problem. By conducting a large number of coupled poromechanical simulations varying the uncertain geomechanical parameters, we yield an exceedance probability of 7%-15% for the Pohang mainshock. Remarkably, this physics-based stochastic prior forecast is quite comparable to the posterior likelihood inferred from the magnitude-frequency relationship of recorded seismicity. Our results reveal a scaling relationship between the earthquake magnitude and the initial fault stability, which indicates a threshold of the initial Coulomb Failure Stress to differentiate if faults are initially, critically stressed, and thus, the earthquake magnitude. This Pohang threshold is −0.2 to −0.1 MPa, about one order of magnitude larger than that proposed for natural earthquakes. This study highlights that the reactivation of critically stressed faults may trigger damaging earthquakes even for small poromechanical perturbations and opens a promising avenue for assessing the likelihood of induced earthquakes based on physical understanding.

The 2017 Pohang, South Korea, $M_\text{w}$ 5.5 earthquake is the largest seismic event induced by any Enhanced Geothermal System (EGS) project. It occurred some two months after the last hydraulic stimulation had been completed[1–3]. Although surface rupture was not observed, other surface manifestations were recorded, such as evident coseismic surface deformations[4,5], ground cracks, and soil liquefactions[6,7]. The epicenter of the earthquake was located near the EGS project site[1,2,8,9], with the hypocenter corresponding to the depth of the injection wells[1,2]. It is now commonly accepted that the earthquake was induced by high-pressure hydraulic stimulation in the well PX-2, which activated a pre-existing low-permeability fault[1,2,10–12]. Yet, the specifics regarding the triggering mechanisms remain the subject of scientific debate and opposing hypotheses have been formulated.

The overseas research advisory committee concluded that the overpressure caused by fluid injection reactivated a pre-existing critically stressed fault, which, in turn, released the stored elastic strain energy, leading to the Pohang earthquake[2,3]. Yeo et al.[9] proposed that the earthquake interactions, i.e., the static stress transfer from the previous induced seismicity, made a greater contribution on the earthquake nucleation than the overpressure. Several researchers[13–16] argued that the poroelastic stress played a notable role in triggering the mainshock, especially enhanced by the low-permeability fault, which requires coupled hydromechanical (HM) approaches to assess the triggering mechanisms. Some of these HM assessments attributed this typical post-injection seismicity to slow fluid diffusion and its subsequent HM response because of the low hydraulic

[1]Department of Civil and Environmental Engineering (DECA), Universitat Politècnica de Catalunya (UPC), Barcelona, Spain. [2]Associated Unit: Hydrogeology Group (UPC-CSIC), Barcelona, Spain. [3]Global Change Research Group (GCRG), IMEDEA, CSIC-UIB, Esporles, Spain. [4]Institute of Environmental Assessment and Water Research (IDAEA), Spanish National Research Council (CSIC), Barcelona, Spain. [5]Geo-Energie Suisse AG, Zürich, Switzerland. [6]Present address: the Institute of Marine Sciences (ICM), Spanish National Research Council (CSIC), Barcelona, Spain. ✉e-mail: haiqing.wu@csic.es

diffusivity of the granitic formation[14,16]. Geochemical effects in terms of fault corrosion have also been proposed to explain the weakening process of the fault[17]. Thus, multiple triggering mechanisms driven by a combination of coupled thermo-hydro-mechanical-chemical processes, similar to those identified at the Castor underground gas storage project site[18,19], likely induced the $M_w$ 5.5 earthquake.

The aforementioned studies on the Pohang earthquake present three limitations. First, all of them assumed that the reactivated fault was initially very close to failure, i.e., critically stressed. Any perturbation larger than 0.01 MPa, the generally assumed Coulomb threshold for natural earthquakes[20,21], may induce the mainshock, with critical estimates ranging from 0.05 to 0.3 MPa[2,3,9,13–15]. However, the overpressure in the source region of the mainshock inferred from focal mechanism solutions is one order of magnitude larger than the previous estimates[22], implying that the fault could not have been so critically stressed prior to stimulation. Otherwise, tidal stress variations (0.001–0.01 MPa)[23] could have triggered similar damaging events. Second, most of the hydraulic and HM assessments focused only on fault stability[2,3,9,13–16], while no scaling relationships between the final evaluated results (pore pressure buildup, transferred Coulomb static stress, poroelastic stress, and Coulomb failure stress) and the earthquake magnitude have been established. Thus, only qualitative knowledge on the triggering mechanisms is available, but quantitative assessments of the earthquake magnitude and the possibility of its occurrence have not been attempted to date. Third, the in-situ stress and fault orientation are uncertain, as illustrated by the range of likely estimates (Tables 1 and 2). Mechanical and HM stability assessments are typically based on one individual estimate, and this has been also the case at Pohang[13–16]. All these evidences show that there was a wide uncertainty on the in-situ stress, fault orientation and rock properties, which control the stability of the fault[24,25]. Therefore, we argue that quantitative estimates of the induced seismicity risk at the Pohang EGS site require a formal uncertainty analysis, rather than a traditional "single best" deterministic analysis[13–16].

The Monte Carlo method is a basic stochastic approach for uncertainty analysis[26,27] that has been applied to various geological hazard assessments. Original applications used the Monte Carlo method to analyze the earthquake recurrence parameters, with varying degrees of physical understanding[28–30]. More recent, physics based, studies have aimed at estimating in-situ stress and rock properties according to field and laboratory data[31,32], or at assessing the impact of spatial and/or temporal variability of model parameters and loads[33–35]. But only Masoudian et al.[33] carried out a formal HM risk assessment of hill slope stability. And their analysis was limited by the relatively small number of realizations and the coarse grid they had to use because of the computational burden. Numerically evaluating the coupled HM process for every Monte Carlo realization is indeed a technical challenge for current computational capabilities. We propose to circumvent such challenge by using verified analytical solutions to accelerate HM evaluations, making stochastic poromechanical analysis (SPA) computationally tractable.

This study aims at developing a SPA approach to evaluate the risk of induced seismicity. This SPA method incorporates poromechanical physics into the Monte Carlo simulations, which provide a way to estimate a priori the probability of an earthquake exceeding a specific magnitude. The approach is described in detail in the Methods section. Since seismicity is sensitive to the stress state, we first make a comparative analysis on the available estimates of the in-situ stress to constrain its plausible range. Second, we do a parametric space analysis to identify which geomechanical properties, both geological and mechanical, play an essential role on initial fault stability. Third, we carry out a deterministic analysis to show the capability of the analytical solutions in representing the poromechanical response and the necessity of uncertainty/stochastic analysis. Finally, we perform Monte Carlo simulations that acknowledge the uncertainty of multiple variables defining the poromechanical problem, and statistically estimate the prior probability of inducing the Pohang earthquake, which yields a quite high value for decision-marking. Results highlight that the reactivation of critically stressed faults may trigger damaging earthquakes even for small poromechanical perturbations.

## Results

### Two oblique-slip patterns of the rupture fault linked to the Pohang earthquake

The current three dimensional (3D) in-situ stress state for the Pohang EGS site at the depth of 4.2 km is either a reverse or a strike-slip faulting regime (Table 2). Projecting the 3D principal stress tensor, i.e., the vertical ($\sigma_v$), maximum ($\sigma_H$) and minimum ($\sigma_h$) horizontal stress components, into the two-dimensional (2D) fault plane reveals that the shear stress acting on the fault plane aligns with neither the dip nor the strike directions, but forms an angle with both regardless of the stress regime (Fig. 1A). An oblique slip occurs in such loading when it goes to failure, with a displacement vector of the hanging wall directed upwards and towards North-East (Fig. 1B). The primary component of this oblique slip is a reverse slip for reverse faulting regimes, and a strike slip for strike-slip faulting regimes, corresponding to the likely two-type estimates of the fault described in Table 1: the dip-slip fault (FP1, FP3 and FP5) and the strike-slip fault (FP7 and FP8). This implies that there are two possible oblique-slip patterns at Pohang: (1) a reverse-slip-dominated pattern accompanied with a minor strike-slip component (RS-S pattern), which involves a reverse fault being in a reverse faulting regime; and (2) a strike-slip-dominated pattern accompanied with a minor reverse component (SS-R pattern), which considers a strike-slip fault being in a strike-slip faulting regime.

### Plausibility of prior estimates of the in-situ stress state

The six potential estimates (IS1-IS6 in Table 2) of the in-situ stress state present a wide range of proposed values. To constrain the plausible range of the in-situ stress, we assess the initial stability of the likely fault orientations (Table 1), based on the linear Mohr-Coulomb criterion, for each of these stress estimates. In particular, we compute the shear and effective normal stress components acting on the inclined fault plane by the stress transformation approach described in the Methods section, and also the initial Coulomb Failure Stress ($CFS^0$). As it turns out, assuming a static friction coefficient $f_{st}$ of 0.5[36], all likely estimates of the fault plane are super-critically stressed in stress states IS1, IS2, and IS6, while are far from critically stressed in IS3, IS4, and IS5 (Fig. 2). This result indicates that none of these stress estimates can perfectly depict the initial state of the active fault, which should be (close to) critically stressed. To make the likely fault planes being initially stable, a highly unlikely value of $f_{st}$ larger than 0.87 is required in IS6 (Fig. 2F). This low likelihood of IS6 may be attributed to the fact that it was estimated from focal mechanism data over the whole Korean Peninsula (Table 2). In contrast, a more plausible $f_{st} \sim 0.61$ is sufficient for such purpose in IS1 and IS2 (Fig. 2A, B). Similarly, $f_{st} \sim 0.45$ could lead the crust to be critically stressed, while only one likely fault plane (FP5) is critically oriented in IS3 and IS5, and no one in IS4 (Fig. 2C–E). Although this stability analysis cannot identify the true stress estimate, it shows a potential range of the in-situ stress for both oblique-slip patterns described above. In particular, the

## Table 1 | Estimates of the fault plane related to the induced seismicity at the Pohang EGS site[2,4]

| Number | Strike (°) | Dip $\theta$ (°) | Physical description |
|--------|-----------|----------|---------------------|
| FP1 | 214 | 51 | Northwest-dipping nodal plane (FP1) and its alternative one (FP2) of the mainshock focal mechanism from first-motion analysis |
| FP2 | 343 | 52 | |
| FP3 | 215 | 58 | Northwest-dipping nodal plane (FP3) and its alternative one (FP4) of $M_w$ 3.2 event |
| FP4 | 339 | 48 | |
| FP5 | 214 | 43 | Plane of PX−2 seismicity |
| FP6 | 180 | 62 | Plane of PX−1 seismicity |
| FP7 | 225 | 75 | Mainshock fault plane from InSAR analysis (FP7), and from moment tensor analysis (FP8) |
| FP8 | 221 | 66 | |

**Table 2 | Estimates of the in-situ stress state at the depth of ~4.2 km for the Pohang EGS site**

| Num. | Azimuth of $\sigma_H$ (°) | $\sigma_h$ (MPa) | $\sigma_h$ (MPa) | $\sigma_v$ (MPa) | Regime | Rock type | $DSR_{max}$ | Reference |
|---|---|---|---|---|---|---|---|---|
| IS1 | N77 ± 23 | 243 | 120 | 106 | RF | Crys | 0.517 | 3 |
| IS2 | N74 | 203 | 93 | 106 | SS | Crys | 0.519 | |
| IS3 | N100 | 198 | 107 | 107 | SS/RF | Crys | 0.412 | 58 |
| IS4 | N100 | 168 | 95 | 107 | SS | Crys | 0.408 | |
| IS5 | N111 | 200 | 120 | 110 | RF | Crys | 0.398 | 17 |
| IS6 | N75 | 256 | 87 | 111 | SS | Crys | 0.653 | 59 |
| IS7 | N130-136E | 138 | 86 | 107 | SS | Sedi | 0.37 | 60,61 |
| IS8 | N65-130E | 115-138 | 81-105 | 110 | SS | Sedi | ≤0.357 | 56 |
| IS9 | N100E | 133-153 | 89-119 | 107 | SS/RF | Crys &Sedi | ≤0.329 | 8 |

RF and SS mean the reverse faulting and strike-slip faulting regimes, respectively. "Sedi" and "Crys" denote that the stress state was estimated from data measured at the sedimentary and crystalline rock, respectively. $DSR_{max}$ is the maximum deviatoric stress ratio (Eq. (1)).

horizontal principal stress and azimuth may range from the value of IS3 to IS1 for the RS-S pattern, and from the value of IS4 to IS2 for the SS-R pattern (Supplementary Table S1).

The likely fault planes FP1, FP3 and FP5 are more critical than FP7 and FP8 in stress states IS1, IS3, and IS5, while the opposite is true for IS2 and IS4 (Fig. 2). This is because faults are prone to slip with a low dip angle (≤(π/2-atan $f_{st}$)/2 = 29.5° for $f_{st}$ = 0.6) in the reverse faulting regime (IS1, IS3, and IS5), and with a high dip angle (close to vertical) in the strike-slip faulting regime (IS2 and IS4) (Supplementary Fig. S1)[24,25,37]. This observation indicates that FP1, FP3 and FP5 may relate only to the RS-S pattern, while FP7 and FP8 only to the SS-R pattern, providing the potential range of fault orientation for each pattern (Supplementary Table S1).

**Effect of geomechanical properties on initial fault stability**

We adopt the mean value of the in-situ stress and fault orientation (Supplementary Table S1) constrained from the above plausibility analysis as a base scenario of these geomechanical properties of the fault, and then consider parametric variations of ±10% around the mean of $\sigma_H$, $\sigma_h$, azimuth, and θ to do a parametric space analysis for both slip patterns. The static friction coefficient $f_{st}$ is also variable in this analysis, with the laboratory test value being its mean. This analysis aims at explicitly showing how geomechanical properties affect the initial stability (represented by $CFS^0$) of the fault at Pohang (Fig. 3). The assumed 10% uncertainty just covers the plausible range of the in-situ stress and fault orientation listed in Supplementary Table S1. Within this parameter interval, the $CFS^0$ is proportional to $\sigma_H$ while inversely proportional to $f_{st}$, $\sigma_h$, azimuth, and θ, for both slip patterns. The steeper the line, the larger the effect of the corresponding parameter. Thus, $\sigma_H$ and $f_{st}$ exert stronger controls on fault stability than the other three properties for the RS-S pattern. In contrast, for the SS-R pattern, the effect of fault dip is neglectable with respect to the others four. Apart from the difference in fault dip between the two slip patterns, the effect of $\sigma_h$ is substantially larger in the SS-R pattern than the RS-S pattern because $\sigma_h$ is the least principal stress in the SS-R pattern while is the intermediate principal stress in the RS-S pattern. It is well-known that the intermediate principal stress has a smaller effect on potential shear failure of rocks than the largest and smallest principal stresses[24,25]. In conclusion, the magnitude of in-situ stress and the static friction coefficient play a more evident role on fault stability than the fault orientation.

**Capability of analytical solutions in assessing induced seismicity potential**

We now present how the fault stability changes after fluid injection using the previous base scenario of the two models (slip patterns) with the mean value of these geomechanical properties to explain the model and the procedure followed in each Monte Carlo simulation. We use the injection data to simulate the evolution of pore pressure changes in the geothermal reservoir (Eq. (2)). The three hydraulic stimulations in PX-2 are simplified as five continuous injection periods (Supplementary Table S2), i.e., a typical

stepwise injection with five cycles. We match the observed pore pressure changes at the bottom hole with the recorded data by adjusting the reservoir thickness, as the other hydraulic parameters can be referred from the literature (Supplementary Table S3) including a reservoir permeability of $5 \times 10^{-18}$ $m^2$. Though the analyzed fault core is assumed to be impermeable, the injection-induced overpressure can still dissipate through the highly permeable fault damage zone[2,3,9] along the longitudinal direction (see Methods section). This differentiated fault structure allows us to assume that no more additional overpressure generates in the injection side of the fault and that no pressure perturbation occurs in the other side.

A reservoir thickness of ~750 m yields a relatively good reproduction of the measured evolution of overpressure (Supplementary Fig. S2). With this thickness, we calculate the radial distribution of pore pressure changes at the instant of the mainshock (Fig. 4). The radius of influence $R_{max}$ is ~586 m in this scenario and the pore pressure change at such location attenuates to ~0.01 MPa. The total length of the pressurized region is the sum of $R_{max}$ and $L_D$ (the distance between the fault and well PX-2, see Fig. 1C) that is 394 m for the RS-S pattern and 169 m for the SS-R pattern using the base scenario of fault dip (Supplementary Table S1). Accordingly, the length of the pressurized region is ~980 m for the RS-S pattern and ~755 m for the SS-R pattern. This analytical representation yields a pressurized region that (1) expands isotropically from the source point, and (2) aligns well with existing numerical simulations[3,9,15]. The mean pore pressure change $\Delta p_m$ in this pressurized region (Eq. (5)) equals 1.85 and 1.97 MPa for the RS-S and SS-R patterns, respectively, and pore pressure changes at the fault ($\Delta p_{FP}$) are 0.06 and 0.7 MPa, respectively, which are consistent with the numerical solutions of Ellsworth et al.[2], Lim et al.[14], Yeo et al.[9] and Wassing et al.[15]. The pressure changes near the bottom hole are also similar to those reported in Wassing et al.[15].

We then use the analytical solution (Eqs. (6)-(8)) to evaluate the poroelastic stress along the fault with the mean overpressure $\Delta p_m$ of the pressurized region, and assess fault stability changes ($\Delta CFS$, Eq. (18)) with the overpressure at the fault $\Delta p_{FP}$ (Fig. 5, see Methods section). Both shear (Fig. 5A) and normal (Fig. 5B) components of the poroelastic stress tend to stabilize the fault patch around P1, while destabilize the fault patch around P2 regardless of the slip pattern (see the definition of $\Delta CFS$ in Eq. (18)) where P1 and P2 are the top and bottom cross-points between the fault and the pressurized region, respectively (Fig. 1C). As a consequence, the fault patch around P2 becomes less stable, with the most weakening point (P2) being located at 4.65 km depth (Fig. 5C), which is consistent with the hypocenter (4.5 km) of the mainshock[1]. Such less stable fault patch also correlates well with the hypocenter of the main foreshocks and aftershocks that occurred within a few hours before or after the mainshock[1-3] (Fig. 5C). This strong spatial agreement verifies the robustness of our modeling approaches and results.

The less stable fault patch undergoes a larger (though local) decrease of fault stability for the RS-S pattern than for the SS-R pattern. Fault dip controls such stability changes, with a higher dip angle (70.5° in the SS-R

**Fig. 1 | Sketch of the conceptual model for the Pohang EGS site at depth. a** Three-dimensional (3D) geological model including the fault plane, the injection well PX−2, and the principal stress tensor ($\sigma_v$, $\sigma_H$, $\sigma_h$) at the in-situ state. **b** Oblique slip along the fault plane resulting from the stress state. **c** Two-dimensional (2D) cross section perpendicular to the fault strike. The normal and shear stress components acting on the vertical plane ($\sigma_{n1}$ and $\tau_1$, respectively) are transformed from $\sigma_H$ and $\sigma_h$, and those acting on the inclined fault plane ($\sigma_{n2}$ and $\tau_2$, respectively) are transformed from $\sigma_{n1}$ and $\sigma_v$ (see Methods section). The module between $\tau_2$ and the projection of $\tau_1$ on the fault plane is the total shear stress. The blue $\oplus$ means the inward direction of the out-of-plane in (**c**). The injection-induced pore pressure and stress changes are shaded in blue and orange, respectively, in (**c**). P1 and P2 are the top and bottom cross-points between the fault and the pressurized region, respectively. $L_D$ is the distance between the fault and the center of the openhole section of the well PX−2, which varies with the fault dip $\theta$.

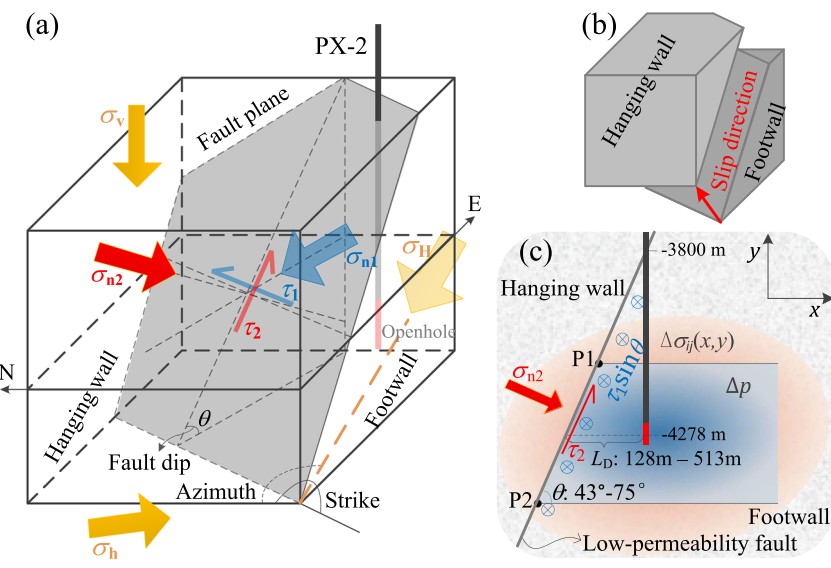

pattern) leading to a smaller absolute variation in shear stress ($|\tau^f| - |\tau^0|$) after stress transformation than a low dip angle (50.5° in the RS-S pattern) (Fig. 5A). In contrast, the central section of the fault patch intersecting the pressurized region becomes more destabilized for the SS-R pattern than for the RS-S pattern, because the overpressure at the fault (Fig. 4), $\Delta p_{FP}$, is one order of magnitude larger in the former than in the latter.

The $CFS^0$ of the fault is -2 MPa for both slip patterns in this mean-value example (Fig. 3). Adding the $CFS^0$ to $\Delta CFS$ yields the final state of the fault at the instant of the mainshock, which remains stable (negative value) because the initial state is far away from failure conditions. When the initial state is more critical, for instance, when $CFS^0$ equals -0.1 MPa, the fault becomes unstable and we can further calculate the size of the slipping fault patch (Eq. (19)) and the magnitude of related earthquakes (Eq. (21)). This example shows that the analytical solutions applied in this work are capable of representing the poromechanical response due to fluid injection (Figs. 4 and 5), while the final stable state of the fault under such deterministic conditions, i.e., the mean value of geomechanical properties, contradicts the observed occurrence of the mainshock. Thus, a stochastic analysis is required to account for the wide uncertainty of the initial geomechanical conditions of the fault.

**Probability distribution of induced seismicity**
We perform an SPA to assess the prior probability of the Pohang earthquake acknowledging the uncertainty of in-situ stress, fault orientation, and rock properties (see Methods section). The RS-S and SS-R cases are examined separately. For each analysis, we simulate 30,000 valid Monte Carlo realizations. The evolving stack of earthquake magnitudes $M_w$ converges after ~25,000 realizations (Supplementary Fig. S3). Importantly, though discarding numerous invalid samples during simulations, the ensemble of valid samples of every random variable still follows the prescribed normal distribution (Supplementary Fig. S4), validating both the robustness of our Monte Carlo simulations and the self-consistency of the distributions assumed for random variables.

Figure 6 displays the probability distribution of the earthquake magnitude $M_w$ conditioned to the pore pressure changes displayed in Fig. 4 and the $CFS^0$ of the fault being within the range of -1 to -0.01 MPa. The lower bound of $CFS^0$ reflects the previous mean-value scenario and the focal mechanism constraints[22], whereas the upper bound corresponds to the Coulomb stress threshold typically associated with natural earthquakes[20,21]. The maximum possible magnitude of induced seismicity is close to $M_w$ 7.0 for both slip patterns, while a substantial fraction of realizations generates

seismic events too small to be detected, i.e., $M_w$ -1.0. The probability density function (PDF) of $M_w$ is approximately uniform with small oscillations for both slip patterns when $M_w$ ranges from -1 to 5, with the RS-S pattern producing slightly higher probability densities than the SS-R pattern (Fig. 6A). For a larger size of earthquakes, the PDFs diverge in character. The RS-S pattern exhibits a monotonic attenuation, whereas the SS-R pattern displays a sharp rise followed by attenuation, and yields higher probability densities than the RS-S pattern when $M_w > 5.5$. These features of PDF are attributed to the spatial distribution of $\Delta CFS$ along the fault, in which the RS-S pattern is more easily initiated than the SS-R pattern while the latter growths faster than the former once arriving the asymptotic failure (Fig. 5C). As a result, the exceedance probability of inducing a specific magnitude of earthquakes is higher for the RS-S pattern than for the SS-R pattern at magnitudes less than $M_w$ 3.5, while becomes opposite for larger magnitudes of events (Fig. 6B). As examples, the exceedance probability of inducing earthquakes with $M_w > 3.0$ and $M_w > 5.5$ is ~34% and ~6.9% for the RS-S pattern, respectively, and ~32% and ~15% for the SS-R pattern, respectively.

**A posteriori analysis of the initial critical state of the fault**
Every Monte Carlo realization represents an initial stress condition of the fault and the poromechanical perturbation. Mapping these realizations into the parametric space with respect to the initial fault understress (-$CFS^0$) and the maximum change in fault stability ($\Delta CFS$ at the cross-point P2) allows us to ascertain which conditions may induce earthquakes at Pohang and, reversely, to infer how critically stressed was the reactivated fault that induced the mainshock. Such space is divided into two regions by a marginally perturbed boundary ($\Delta CFS = -CFS^0$), below which the fault remains stable at the instant of the mainshock (Fig. 7). Regardless of the slip pattern, all realizations cluster at the region representing fault instability as a result of poromechanical perturbation, in which the maximum change in fault stability is larger than the initial fault understress. A part of realizations depicts the limiting case of initially critically stressed fault, leading to damaging earthquakes. Another part of realizations depicts the limiting case of marginally perturbed fault, which results in microseismicity. The primary distinction between the two slip patterns is that the maximum stability change for the RS-S pattern is larger than that attained for the SS-R pattern, like that illustrated in Fig. 5C. This difference arises from the influence of fault dip on stress transformation, which distinctly affects the size of the poromechanical stress acting on the fault (Fig. 5A).

Plotting the earthquake magnitude $M_w$ as a function of the $CFS^0$ reveals a scaling relationship between them, which is linear when $M_w < 5.3$ and

**Fig. 2 | Initial stability of likely estimates of the fault at Pohang.** Mohr-Coulomb diagram is calculated with the in-situ stress states **a** IS1, **b** IS2, **c** IS3, **d** IS4, **e** IS5, and **f** IS6. The black lines are Mohr-Coulomb failure envelopes under different values, as indicated in the legends, of the static friction coefficient $f_{st}$, in which its laboratory test value is indicated by the solid lines, and the others (indicated by dashed lines) are derived by making the crust just being critically stressed. The dots with numbers denote the five likely estimates of the fault plane (FP1, 3, 5, 7 and 8; Table 1), which are located by the solved shear and effective normal stress components and colored by the initial Coulomb failure stress ($CFS^0$), with the uniform color-scale legend being included in (**d**).

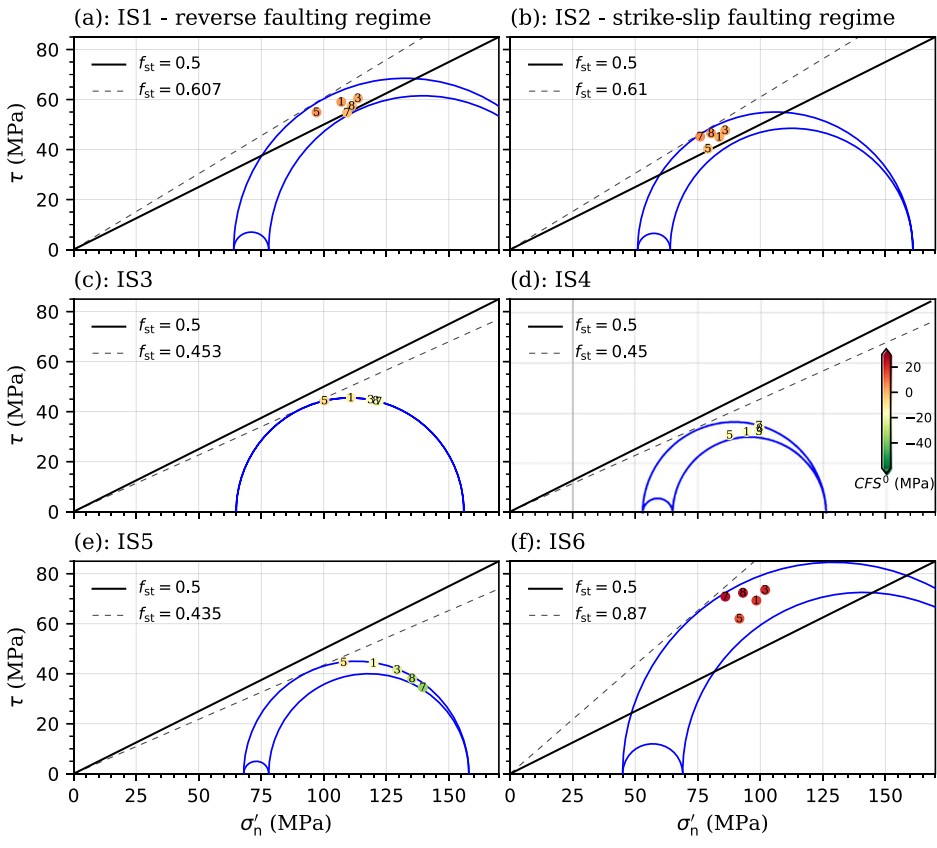

**Fig. 3 | Parametric space analysis.** The initial fault stability ($CFS^0$) as a function of the relative change (± 10%) with respect to the mean of several geomechanical properties ($\sigma_H$, $\sigma_h$, azimuth, $\theta$, and $f_{st}$) for **a** the RS-S and **b** SS-R patterns. The cross-point of all curves corresponds to the result evaluated with the means, considered here as the base geomechanical characterization of the fault.

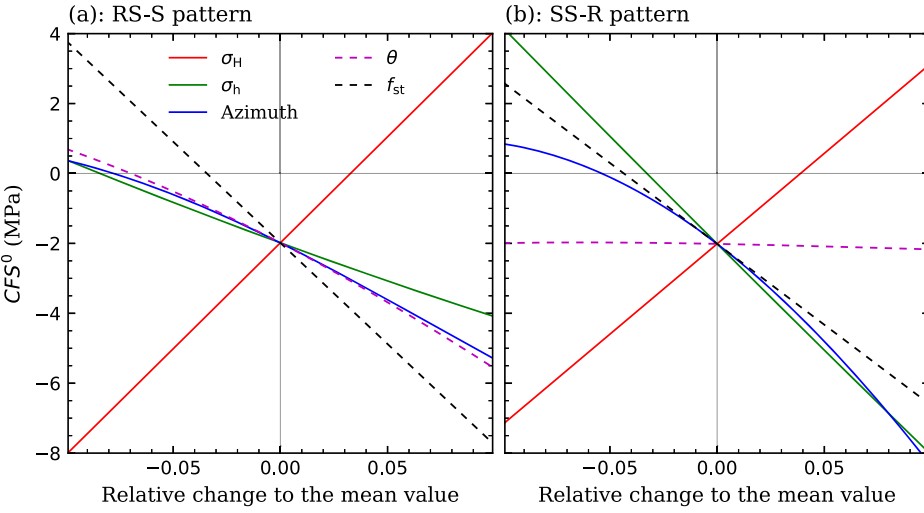

transforms to be exponential when $M_w > 5.3$ (Fig. 8A). The transition point is quite similar between the two slip patterns, although their scaling coefficients differ. Moreover, the linear scaling coefficient varies with the value of $\Delta CFS$, whereas $\Delta CFS$ hardly has effect on the exponential scaling coefficient. A similar effect of $\Delta CFS$ is observed in the correlation of $M_w$ and $\Delta CFS$, in which their scaling changes from logarithmic to linear, and finally no relevant with increasing the value of $CFS^0$, i.e., decreasing the initial fault stability (Fig. 8B). These trends indicate a threshold of $CFS^0$ that differentiates if faults are critically stressed under initial conditions, which is about -0.1 MPa for the RS-S pattern and -0.2 MPa for the SS-R pattern at Pohang (Figs. 7 and 8). For such a critically stressed fault, a decrease in fault stability exceeding 0.5 MPa may trigger a damaging earthquake whose magnitude is

dominated by the fault understress, i.e., the earthquake energy mainly results from the release of the tectonic strain. In contrast, both the fault understress and poromechanical perturbation affect the magnitude of induced earthquakes for less critically stressed faults. The less critically stressed the fault, the larger the impact of the poromechanical perturbation on the earthquake magnitude.

## Discussion

Two oblique-slip patterns (RS-S and SS-R) are found plausible for analyzing triggering processes of the Pohang earthquake (Fig. 1). Both are consistent with observed ground deformation[6]. Fault stability analysis under initial stress conditions identifies the plausible range of the in-situ stress state and

fault orientation for both slip patterns (Fig. 2). The less stable fault patch evaluated for both slip patterns (Fig. 5C) agrees well with the hypocenter of the foreshocks, mainshock, and most of the immediate aftershocks[1–3], supporting the verification of our model and results. Fault reactivation is more prone to be initiated with the RS-S pattern than with the SS-R pattern as a result of poromechanical perturbation, despite the overpressure at the fault being one order of magnitude smaller in the former than in the latter (Figs. 4 and 5). However, the slipping fault patch size growths faster for the SS-R pattern than for the RS-S pattern (Fig. 5C). As a consequence, the probability of inducing small earthquakes is higher for the RS-S pattern than for the SS-R pattern, while the opposite is true for damaging earthquakes (Fig. 6). For the Pohang $M_w$ 5.5 earthquake, our physics-based stochastic simulations forecast a prior probability of 7% for the RS-S pattern and of 15% for the SS-R pattern (Fig. 6), being quite comparable to the posterior estimate inferred from the magnitude-frequency relationship of recorded seismicity that is about 6% with a *b*-value of 0.73 and 16% with a *b*-value of 0.65[39]. Thus, the RS-S pattern correlates with a high *b*-value, whereas the SS-R pattern a lower *b*-value. Such comparable results, on the one hand, verify

our SPA method and, on the other hand, highlight the need for additional constraints regarding site characterization to further determine the more likely slip pattern at Pohang[13,38], as both patterns can reproduce well the recorded seismicity with a reasonable *b*-value.

Existing estimates of the in-situ stress tensor (Table 2) were obtained using an assumed static friction coefficient $f_{st}$. For instance, IS1 and IS2 were estimated with a value of 0.6[2,3]. Thus, plausibility analysis on these estimates shows that the fault is either supercritical or subcritical when adopting the test value of 0.5[36] (Fig. 2). The fault is initially too stable ($CFS^0$ = -2) even evaluating with the mean value of several these stress estimates (Fig. 3). In our stochastic simulations, the valid Monte Carlo realizations inversely indicate two new estimates that is 222 ± 17 MPa for $\sigma_H$, 113.4 ± 6.5 MPa for $\sigma_h$, and 92.6 ± 15° for the azimuth when considering the RS-S pattern and 186.9 ± 14 MPa for $\sigma_H$, 94 ± 1 MPa for $\sigma_h$, and 84.9 ± 10° for the azimuth when considering the SS-R pattern (Supplementary Fig. S4). The initial stability ($CFS^0$) of the fault is -1.1 MPa for the RS-S pattern and -0.6 MPa for the SS-R pattern with such new mean estimates, which is more reasonable for the Pohang EGS site given the occurrence of induced seismicity.

The stochastic results reveal a scaling relationship between the earthquake magnitude and initial fault stability, which is linear for small to moderate earthquakes and exponential for damaging earthquakes (e.g., $M_w$ > 5.0, Figs. 7 and 8). Such linear and exponential scaling relationships correspond to the initiation and asymptotic failure stages of fault slip, respectively (Fig. 5C). This behavior demonstrates a threshold of $CFS^0$ that can differentiate critically stressed faults from marginally stressed ones at initial state, which is about -0.2 to -0.1 MPa for the Pohang EGS site (Figs. 7 and 8), being one order of magnitude larger than the threshold proposed for natural earthquakes (about -0.01 MPa)[20,21]. This finding implies that the fault does not need to be as close to failure as assumed in previous studies[2,3,9,13–15]. Otherwise, earthquakes of $M_w$7.0 might have occurred at Pohang. Whether this site-dependent threshold is valid for other geo-energy project sites requires further research that is beyond the scope of this contribution. However, the methodology proposed here is generic, reproducible, and applicable to any site.

Our results also indicate that the poromechanical response dominates the fluid-induced reactivation of non/sub-critically stressed faults, which usually lead to microseismicity. In contrast, such response only initiates the reactivation of critically stressed faults, which then turns to asymptotic failure and the release of tectonic strain supports the fault rupture, triggering

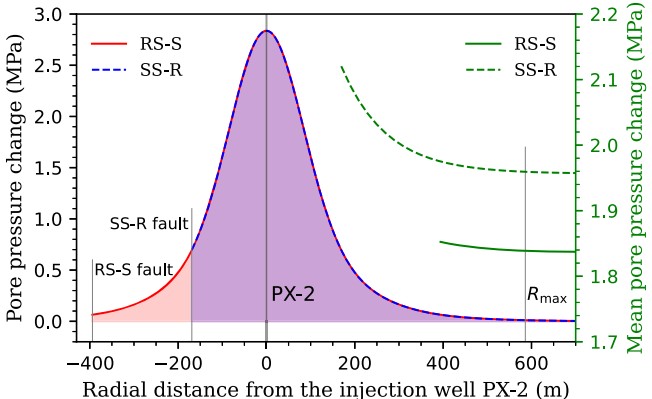

**Fig. 4 | Hydraulic modeling.** Spatial distribution of pore pressure changes at the instant of the mainshock in the base case scenario for both RS-S and SS-R patterns. The mean pore pressure change of the pressurized region as a function of the radial distance is also included (green axis). The location of RS-S and SS-R faults corresponds to fault dips of 50.5° and 70.5°, respectively.

**Fig. 5 | Poroelastic modeling.** Injection-induced **a** shear (Δτ) and **b** normal (Δσ_n) stress variations at the instant of the mainshock, and **c** Coulomb Failure Stress Change (ΔCFS) along the fault plane in the mean-value example for both RS-S and SS-R patterns. A value of 0.8 is adopted for Biot's coefficient to evaluate the poroelastic stress[13,38]. As a comparison, we also plot the difference in absolute shear stress ($|τ^f| − |τ^0|$) in (**a**) and the effective normal stress variations (Δσ'_n) in (**b**). The results along the fault are projected on the vertical *y* axis. The depth of the hypocenters of six foreshocks (cyan dots), mainshock (red star), and 210 aftershocks (black dots) recorded in the first three hours following the mainshock[1] is displaced in (**c**), where only the *y*-coordinate of dots has physical meaning. The depth of injection center is indicated by the horizontal gray line.

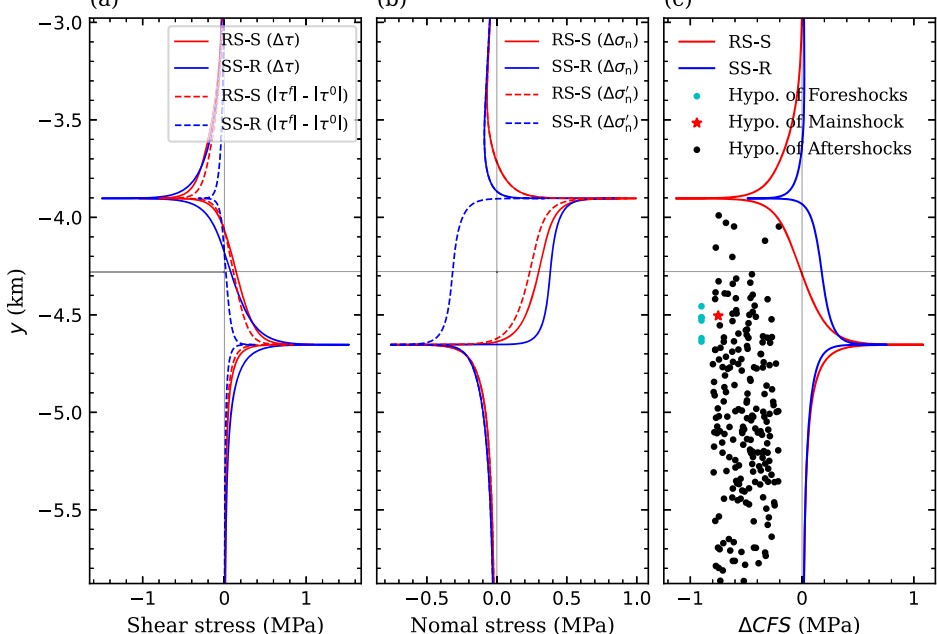

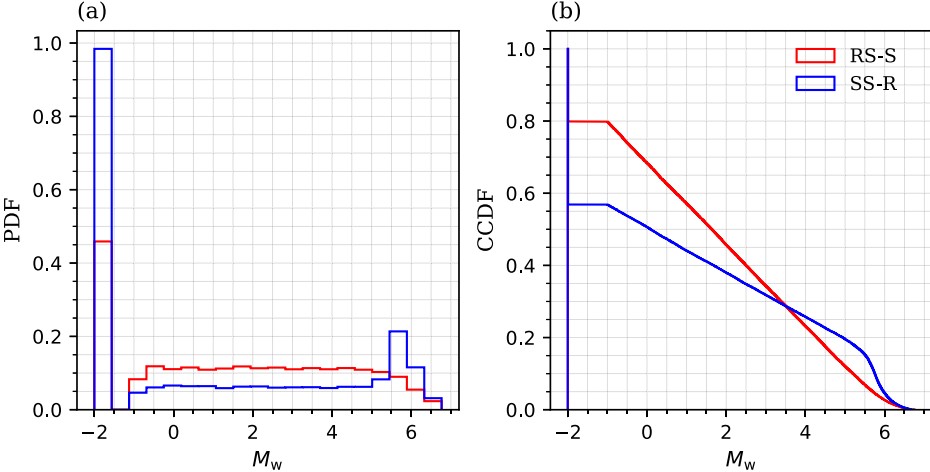

**Fig. 6 | Stochastic poromechanical simulations.** **a** Probability density function (PDF) and **b** complementary cumulative distribution function (CCDF) of the earthquake magnitude ($M_w$) for both RS-S and SS-R patterns.

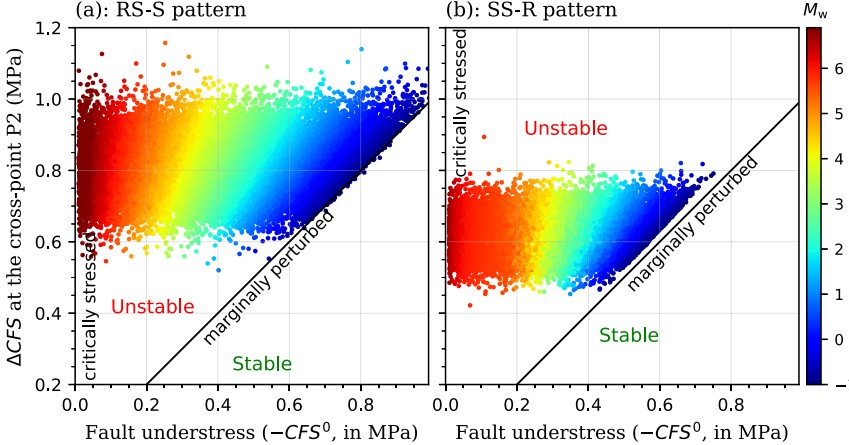

**Fig. 7 | Fault stability and earthquake magnitude map.** Distribution of the Monte Carlo realizations excluding the cases of $M_w = -2$ in the parametric space with respect to the fault understress ($-CFS^0$) and the maximum change in fault stability ($\Delta CFS$ at the cross-point P2) for **a** the RS-S and **b** the SS-R patterns. Each dot depicts a single realization, defined by its $CFS^0$ and $\Delta CFS$ at P2, and colored by its moment magnitude ($M_w$).

damaging earthquakes (Figs. 5 and 7). These observations highlight that, on the one hand, a small poromechanical perturbation capable of activating critically stressed faults may trigger damaging earthquakes; on the other hand, geo-energy projects are unlikely to directly induce damaging earthquakes[40–42], which are usually categorized as triggered seismicity[2,43]. Hence, the Pohang $M_w$ 5.5 earthquake represents a typical triggered seismicity instead of a simple induced one[44]. That explains why existing scaling laws[45–47] fail to link the earthquake magnitude to the total injected volume in such triggered cases[3,12,48]. Pressure management[49] and TLS (Traffic Light System)-based approaches[50–52] may not work effectively for managing triggered seismicity linked to critically stressed faults. In this line of arguments, more efforts are required in site characterization at the prior stage of projects to detect preexisting faults and the stress state[13,19].

The stochastic simulations are conditioned to the $CFS^0$ being within the range of -1 to -0.01 MPa. We determine such condition given the previous deterministic analysis and existing publications. To evaluate the goodness of this condition and whether or not it can represent reality at Pohang, we repeat our Monte Carlo simulations for several different intervals of $CFS^0$. Results show that the probability of microseismicity increases with decreasing the lower bound of the $CFS^0$ interval and the probability of damaging earthquakes decreases with the upper bound for both slip patterns (Fig. 9). This is consistent with the expectation that a more stable initial state implies a smaller size of induced seismicity. Thus, the probability of $M_w$ 5.5 in these comparison cases diverges more from its posterior estimate in contrast to the original case adopted, verifying the reasonableness of the original interval of $CFS^0$. Nonetheless, the related new estimates for the in-situ stress hardly vary with changing the $CFS^0$ interval and the same scaling

relationships and threshold of critically stressed faults identified above remain unchanged in all cases (Supplementary Figs. S5–S14). This sensitivity analysis indicates that such findings represent some intrinsic characteristics of fluid-induced seismicity. Similar analysis can also be performed for different intervals of Biot's coefficient and static friction coefficient, with expectedly similar outcomes.

It is worth to note that this study considers only the pore pressure diffusion and poromechanical effect as triggering mechanisms among the ones that have been proposed for fluid-induced seismicity. Further extensions are suggested to include (1) the slip-weakening effect on the shear frictional strength, which can increase the final size of the slipping fault patch[53,54]; (2) the Coulomb stress transferred from foreshocks[9], especially the $M_w$ 3.2 seismic event that occurred on April 15, 2017, which may have altered fault stability months before the mainshock. In any case, verified analytical solutions are recommended for representing physics while extending this study because it is hardly possible to perform a stochastic coupled poromechanical simulation acknowledging multiple random variables simultaneously for a larger number of realizations with numerical solutions[33–35]. Another possible line of improvements is to acknowledge more complex distributions of parameters than the independence assumed here, as their joint distribution is not known. This limitation may be extended once more progress about the joint distribution of these variables is achieved.

This study proposes a practical approach (SPA) to forecast the prior probability of inducing earthquakes of a given magnitude, expressed through the magnitude probability density function, due to fluid injection activities. The SPA explicitly acknowledges the broad uncertainty of

**Fig. 8 | Quantitative scaling plot of the Monte Carlo simulations.** Earthquake magnitude $M_w$ as a function of **a** the $CFS^0$ and **b** the Coulomb Failure Stress Change ($\Delta CFS$) at the cross-point P2 for both RS-S and SS-R patterns. The dots imply the Monte Carlo realizations that are extracted from Fig. 7 at fixed values of $\Delta CFS$ at P2 in (**a**) and of $CFS^0$ in (**b**). The gray lines indicate the transition from the linear scaling for $M_w < 5.3$ to the exponential scaling for $M_w > 5.3$.

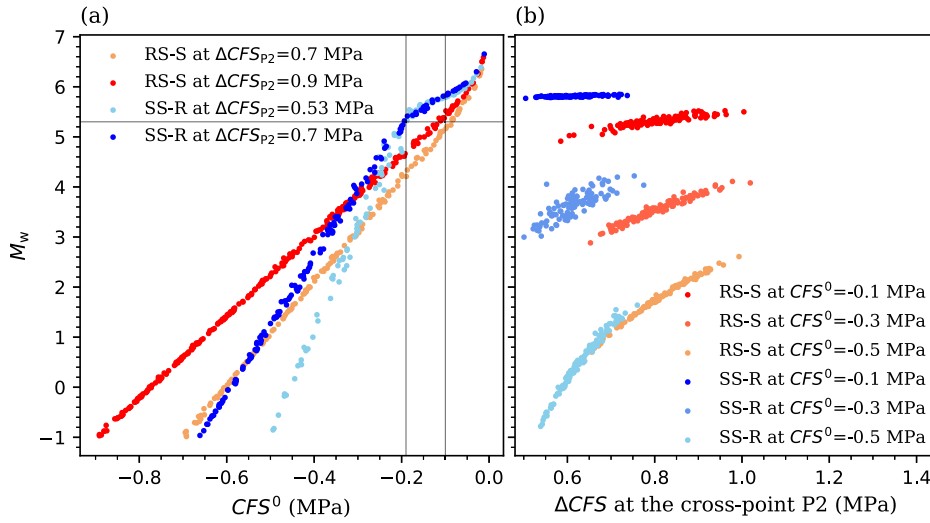

geomechanical properties. Application to the Pohang earthquake yields a 7%–15% prior probability for the mainshock and some 30% for a $M_w$ 3.0 of earthquake at the Pohang fault, which are indeed relevant for decision-making.

We argue that physics-based risk assessments should be required during not only hydraulic stimulations, but also operations. Stimulation itself should be leveraged as an opportunity to improve understanding of the subsurface system[24,25,37] and reduce its inherent uncertainty. The information gathered from monitoring systems should be used to update the model and uncertain parameters, so as to guide operational decisions to control the risk of triggering damaging earthquakes. This vision becomes realistically actionable through tools like the proposed SPA method, especially when integrated with modern data-driven techniques like machine learning. As suggested by Lee[3] and Lee et al.[12], such adaptive, physics-grounded strategies are essential for geothermal projects to succeed in the future. In fact, while several geothermal projects have been suspended due to induced seismicity, the SPA approach is not limited to EGS. Other types of geo-energy projects, such as oil and gas production, subsurface energy storage, and reservoir impoundment, have suffered from the same triggering mechanisms discussed here. Implementing the pioneering SPA framework offers a promising path toward reducing induced seismicity risk and ensuring the sustainable development of these geo-energy projects that are crucial for the future.

## Methods
### Pohang setting
The Pohang EGS project was deployed to produce geothermal energy from a granitic formation around 4.2 km deep, with the strategic goal of mitigating energy deficiency in South Korea[1,3]. Pohang is one of the highest heat-flux areas in South Korea[55]. The project site is located within the Pohang basin that is framed by two major tectonic structures, the Yangsan fault system and the Ulsan fault system[4]. The former involves many N- or NNE-striking strike-slip faults, while the latter includes many typically NNE- to NNW-striking reverse faults[3]. Despite the long tectonic history of these structures, early earthquakes associated with these faults had little effect on fault stability in the area of the EGS project site[2,4]. Five hydraulic stimulations were conducted through the injection wells PX-1 and PX-2, which had been drilled to depths 4215 m and 4348 m, respectively[3]. Several publications have presented the recorded injection data and seismic events, with earthquake magnitudes ranging from $M_w$ -1 to the mainshock of $M_w$ 5.5, recorded on November 15, 2017, at the coastal city of Pohang, South Korea[1–3,9].

Three independent evidences confirm that the mature fault associated with the 2017 Pohang earthquake intersects the well PX-2 at the depth of ~3832 m[2,3,12]. The fault orientation (strike and dip angle $\theta$) is uncertain, and eight estimates (Table 1) have been derived from seismological and geodetic analyses[2,4]. The data-based reversed solutions (FP7 and FP8) depict a northwest-dipping fault[4], excluding the alternative solutions (FP2 and FP4) of focal mechanisms. The fault strikes of FP1, FP3, FP7 and FP8 align closely with that of FP5, which indicates that the mainshock is induced by hydraulic stimulations in PX-2. Thus, FP1, FP3, FP5, FP7 and FP8 are the likely estimates of the fault plane linked to the mainshock (termed here "the fault plane"). Within this subset, FP1, FP3 and FP5 show an intermediate dip angle with a nearly equal strike, while FP7 and FP8 present a higher dip angle with a similar strike, which can be classified as a dip-slip fault and strike-slip fault, respectively[24]. Notably, these estimates show similarity with the previous documented quaternary faulting at the rim of the Pohang basin[3].

Dipole sonic logging of the PX-2 borehole only provided a proper constrain on the vertical stress component $\sigma_v$ (106 MPa at 4.2 km)[3,56,57]. However, the azimuth of the maximum horizontal principal stress $\sigma_H$ showed a ~30% uncertainty because of the anisotropy features[3]. Accordingly, $\sigma_H$ and the minimum horizontal principal stress $\sigma_h$ had to be estimated. The in-situ stress tensors of Table 2 were obtained under different assumptions and methods. IS1 was derived from the logging-while-drilling of PX-2, under the assumption of critically stressed crust characterized by a friction coefficient of 0.6[2,3]. The same assumption was applied to estimate IS2 but using the regional stress attained from the focal mechanism solutions of prior earthquakes between 1997 and 2016 in the vicinity of the Pohang EGS site[2,3]. IS3 and IS4 were modified from IS1 and IS2, respectively, by accounting for spatial or temporal influence of the deep hydraulic stimulations[58]. IS5 and IS6 were deduced from focal mechanism, with IS5 incorporating kinematic indicators of quaternary faults distributed within the Pohang basin[17], and IS6 considering the prior earthquakes over the entire Korean Peninsula[59]. IS7, IS8, and IS9 were estimated from data of the overlying sedimentary rocks[8,56,60,61], which may not adequately represent the stress state at the deeper crystalline basement.

We define a maximum deviatoric stress ratio ($DSR_{max}$) as a first-order metric for evaluating the plausibility of the proposed stress tensors,

$$DSR_{max} = \frac{\sigma_1 - \sigma_3}{\sigma_1 + \sigma_3 - 2p^0},\qquad(1)$$

where $\sigma_1$ and $\sigma_3$ are the greatest and the least principal stress components, respectively, and $p^0$ is the initial pore pressure.

**Fig. 9 | Sensitivity analysis on the initial fault stability.** Complementary cumulative distribution function (CCDF) of the earthquake magnitude ($M_w$) conditioned to several different intervals of the $CFS^0$ for the **a** RS-S and **b** SS-R patterns.

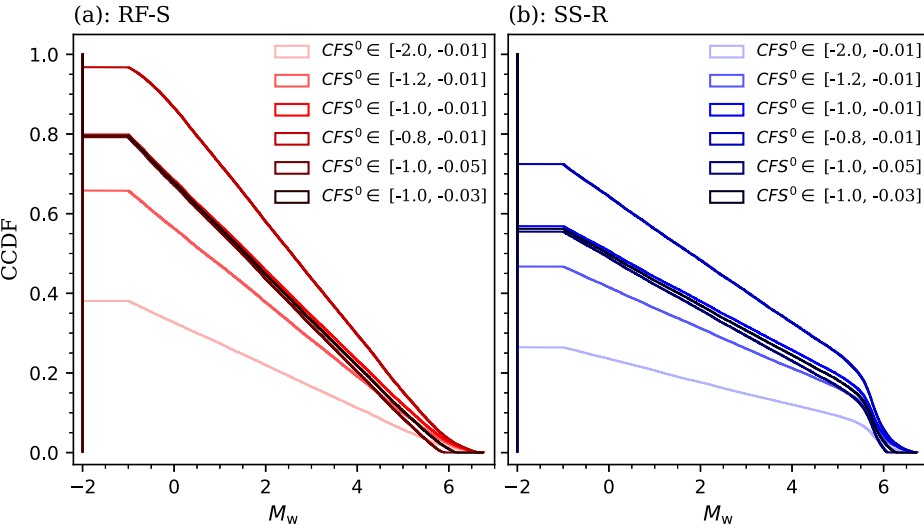

Fault stability is partly controlled by the static friction coefficient $f_{st}$ ($\tan\Theta$, in which $\Theta$ is the friction angle). Laboratory tests on the core samples of PX-2 shows that $\Theta$ is ~26.6°, corresponding to a $f_{st}$ of ~0.5[36,56]. The fault gouge at Pohang was found to contain 10-15% of chlorite[3]. Since friction coefficient decreases with increasing chlorite content[62], reaching values as low as 0.2-0.3 for pure chlorite[63], a friction coefficient of 0.5 (lower than the typical value 0.6) is considered reasonable for the Pohang fault[36]. For a given in-situ stress tensor, the maximum shear plane becomes critically stressed when $DSR_{max}$ reaches $\sin\Theta$ (0.45 at Pohang) according to the 3D linear Mohr circle diagram. Thus, we take 0.45 as a reference value of $DSR_{max}$ for this first-order assessment. Assuming a hydrostatic pore pressure condition[2], estimates IS7, IS8, and IS9 can be excluded because their $DSR_{max}$ values fall distinctly below the reference value (Table 2). Although IS3 to IS5 yield $DSR_{max}$ values slightly smaller than 0.45, we retain them due to the uncertainty in the fault friction angle, which is presently constrained by only two sample test results[36]. In short, we have six potential estimates for the in-situ stress state.

## Analytical evaluation of pore pressure and poroelastic stress changes

The spatiotemporal pore pressure evolution in an infinite homogenous and isotropic aquifer overlain and underlain by impermeable layers is obtained from Theis's solution[64], which, after superposition to represent cyclic injections, reads

$$\Delta p(t) = \sum_{i=1}^{n} \frac{Q_i - Q_{i-1}}{4\pi\lambda h} W(u), \qquad (2)$$

$$u = \frac{\phi c_t r^2}{4\lambda(t - t_{i-1})}, \qquad (3)$$

where $\Delta p(t)$ is the pore pressure change (positive for injection), $t$ is time, $Q_i$ is the volumetric injection rate during the $i$-th injection period ($n$ periods in total), which ends at time $t_i$ ($Q_0 = 0$ m³/s at time $t_0$), $h$ is aquifer thickness, $\lambda$ is mobility defined as the ratio of permeability $k$ to fluid viscosity $\varphi$, $W(u)$ is the Theis well function, $\phi$ is rock porosity, $c_t$ is the bulk compressibility of pore and fluid, and $r$ is the radial distance to the injection well.

Evaluation of the Theis well function usually uses series expansion in the field of groundwater. However, it is accurate for small values of $u$ only[65,66]. For instance, the famous approximation of Cooper and Jacob[67] provides a good estimate for $u < 0.01$ only. The argument $u$ will exceed such

thresholds when the radial distance $r$ reaches a certain value. To tackle this issue, we apply Barry's approximation[65] to evaluate $W(u)$

$$W(u) = \frac{\exp(-u)ln[1 + \frac{G}{u} - \frac{1-G}{(m+bu)^2}]}{G + (1-G)\exp(\frac{-u}{1-G})}, \qquad (4)$$

where $G = \exp(-\gamma)$, $b = \sqrt{\frac{2(1-G)}{G(2-G)}}$, $m = \frac{1}{1+u\sqrt{u}} + \frac{m_\infty q}{1+q}$, $m_\infty = \frac{(1-G)(G^2-6G+12)}{3G(2-G)^2 b}$, $q = \frac{20}{47}u\sqrt{\frac{31}{26}}$, and $\gamma$=0.5772156649 is the Euler constant. Equation (4) is valid across the full range of the argument $u > 0$ and provides sufficient accuracy for data analysis in hydrological applications[65].

Logging data showed that the low-permeability fault core is surrounded by high-permeability damage zones[2,3,9], which implies that the low-permeability fault remains permeable along its longitudinal direction and thus, the induced overpressure can dissipate along the fault. In summary, the extended fault structure in terms of permeability can be simplified as three zones: an impermeable fault core, a highly permeable fault damage zone, and the host rock with in-between permeability[3,9]. When acknowledging such fault structure, we can assume that the transverse blocking effect of the low-permeability fault core is local and that at the model scale some water can flow through singular discontinuities of the fault damage zone, connected by the longitudinal transmissivity. As a result, Theis's solution remains applicable to evaluate the pore pressure changes within the injection side of the fault when pore pressure diffusion reaches the fault, whereas no pressure alters in the other side.

We then apply an analytical solution to evaluate the poroelastic stress induced by pore pressure changes in the reservoir. This solution is derived from the inclusion theory under plane strain conditions[68], and allows quick evaluation of stress variations inside and outside the pressurized region that could be intersected by a permeable or an impermeable fault with an arbitrary fault offset and dip. A 2D cross-section (Fig. 1C) of the 3D overpressure distribution is used to couple with the poroelastic stress solution. The mean pore pressure change $\Delta p_m$, averaged over the 2D pressurized region, is defined by

$$\Delta p_m = \frac{1}{A}\int_A \Delta p dA = \frac{\sum_{i=1}^{l} \Delta p_i \cdot \Delta p_i h dr}{\sum_{i=1}^{l} \Delta p_i h dr}, \qquad (5)$$

where $A$ is the area of the pressurized region that is vertically and equally divided into $l$ sub-regions with each of them being characterized by a geometric width of $dr$ and a pressure-weighted height of $\Delta p_i h$, where $\Delta p_i$ is the overpressure of the $i$-th sub-region. Since Theis's solution extends infinitely in the radial direction, the finite size of the pressurized region is trimmed at the radial distance where $\Delta p_m$ reaches a stationary point, which is defined as the radius of influence $R_{max}$.

The low-permeability fault and the well PX−2 constitute a case of fluid injection into the right-hand side of a low-permeability fault in space (Fig. 1C). Once the pore pressure diffusion reaches the fault, the pressurized region transforms into a trapezoidal inclusion domain, which leads to[68]

$$\Delta\sigma_{xx}(x,y) = -\frac{(1-2\nu)\alpha\Delta p_{\mathrm{m}}}{2\pi(1-\nu)}\left\{ \mathrm{atan}\frac{y-y_2}{x-d} - \mathrm{atan}\frac{y-y_1}{x-d} - \frac{\sin\theta\cos\theta}{2}\ln\frac{f_2(x,y,y_2)}{f_2(x,y,y_1)} - \left[f_1(x,y,y_2)-f_1(x,y,y_1)\right]\sin^2\theta - 2\pi\delta_\Omega \right\} \quad (6)$$

$$\Delta\sigma_{yy}(x,y) = -\Delta\sigma_{xx}(x,y) - \frac{(1-2\nu)\alpha\Delta p_{\mathrm{m}}}{(1-\nu)}\delta_\Omega, \quad (7)$$

$$\Delta\sigma_{xy}(x,y) = -\frac{(1-2\nu)\alpha\Delta p_{\mathrm{m}}}{2\pi(1-\nu)}\left\{ \left[f_1(x,y,y_2)-f_1(x,y,y_1)\right]\sin\theta\cos\theta - \frac{\sin^2\theta}{2}\ln\frac{f_2(x,y,y_2)}{f_2(x,y,y_1)} + \frac{1}{2}\ln\frac{f_3(x-d,y-y_2)}{f_3(x-d,y-y_1)} \right\} \quad (8)$$

where $\Delta\sigma_{ij}$ is the induced stress component (positive for compression) along the direction $j$ and acting over the surface $i$, $x$ and $y$ are the Cartesian coordinates in the vertical plane orthogonal to the fault strike (we denote the axes defining such plane as $x$-$y$ with the origin being located at the well PX-2 and at the ground surface in stress evaluation, see Fig. 1C), $y_1$ and $y_2$ denote the bottom and top boundaries of the trapezoidal inclusion domain $\Omega$, respectively, $d$ is the length of the pressurized region measured at its vertical center for a trapezoidal domain, $\nu$ is Poisson's ratio, $\alpha$ is Biot's coefficient. $\delta_\Omega$ is the modified Kronecker delta, which equals 1 if $(x,y)\in\Omega$ or 0 if $(x,y)\notin\Omega$, and functions $f_1$, $f_2$ and $f_3$ are

$$f_1(x,y,\hat{y}) = \mathrm{atan}\frac{(x-\hat{y}\cot\theta)\cot\theta + (y-\hat{y})}{x - y\cot\theta}, \quad (9)$$

$$f_2(x,y,\hat{y}) = (x-\hat{y}\cot\theta)^2 + (y-\hat{y})^2, \quad (10)$$

$$f_3(x-\hat{x},y-\hat{y}) = (x-\hat{x})^2 + (y-\hat{y})^2. \quad (11)$$

## Stress transformation

We transform the previous poroelastic stress formulated in the $x$-$y$ plane to the fault plane by coordinate transformation. The general stress transformation formula in geomechanics from a given coordinate system $x$-$y$ to another arbitrary coordinate system $x'$-$y'$ is

$$\sigma_{x'x'} = \frac{\sigma_{xx}+\sigma_{yy}}{2} + \frac{\sigma_{xx}-\sigma_{yy}}{2}\cos(2A_{xx'}+\pi) + \sigma_{xy}\sin(2A_{xx'}+\pi), \quad (12)$$

$$\sigma_{x'y'} = -\frac{\sigma_{xx}-\sigma_{yy}}{2}\sin(2A_{xx'}+\pi) + \sigma_{xy}\cos(2A_{xx'}+\pi), \quad (13)$$

where $A_{xx'}+\pi/2$ is the angle between the original axis $x$ and the new axis $x'$. For the geological model shown in Fig. 1C, the normal and shear components ($\Delta\sigma_n$ and $\Delta\tau$) of the poroelastic stress along the fault plane are obtained by setting $A_{xx'}$ to the fault dip $\theta$.

We also apply the previous 2D stress transformation formula to map the 3D in-situ stress tensor (Table 2) onto the 2D fault plane of interest, allowing quantitative assessment of the initial fault state and coupling with the 2D solutions of poromechanical response. This requires two sequential rotations of the coordinate system. We first rotate the two horizontal principal stress components ($\sigma_H$ and $\sigma_h$, corresponding to $\sigma_{xx}$ and $\sigma_{yy}$,

respectively, of Eqs. (12) and (13)) to the vertical plane parallel to the fault strike, which yields $\sigma_{n1}$ and $\tau_1$ (Fig. 1A). This is a rotation with respect to the vertical axis in which $A_{xx'}$ is the angle between the opposite direction of fault strike and the azimuth of $\sigma_H$, i.e., azimuth − (strike − 180°). Second, we transform the evaluated $\sigma_{n1}$ and the vertical principal stress $\sigma_v$ to the inclined fault plane, obtaining $\sigma_{n2}$ and $\tau_2$ (Fig. 1A), which corresponds to a rotation with respect to the fault strike with an angle $A_{xx'}$ equivalent to $\theta$. Thus, the initial normal and shear stress components on the fault plane are

$$\sigma_n^0 = \sigma_{n2} \quad (14)$$

$$\left|\tau^0\right| = \sqrt{(\tau_1\sin\theta)^2 + \tau_2^2} \quad (15)$$

where the superscript 0 represents the initial state, and $\tau_1$ is projected onto the inclined fault plane from its original vertical plane before the module operation.

During and after fluid injection, the induced shear stress $\Delta\tau$ should be added to $\tau_2$ only to render the total shear stress $\tau^f$

$$\left|\tau^f\right| = \sqrt{(\tau_1\sin\theta)^2 + (\tau_2+\Delta\tau)^2} \quad (16)$$

The above stress transformation method offers three key advantages compared with the standard 3D coordinate transformation in matrix form[69,70]: (1) the shear stress direction on the fault plane is explicitly defined; (2) the coupling with the induced in-plane shear stress becomes straightforward; and (3) it avoids the complex 3D geometric operations required to find the angles between the normal direction of the fault plane and the original principal stress tensor.

## Fault stability assessment

Once the normal and shear stress components on the fault plane are specified, we adopt the Coulomb Failure Stress ($CFS$)[21] and Coulomb Failure Stress Change ($\Delta CFS$) to evaluate fault stability and its change, respectively,

$$CFS = \left|\tau^f\right| - f_{\mathrm{st}}\left(\Delta\sigma_n' + \sigma_n'^0\right), \quad (17)$$

$$\Delta CFS = \left|\tau^f\right| - \left|\tau^0\right| - f_{\mathrm{st}}\Delta\sigma_n', \quad (18)$$

where $f_{\mathrm{st}}$ is the static friction coefficient, and the normal stress with a superscript ′ means the effective normal stress, i.e., $\sigma_n'^0 = \sigma_n^0 - p^0$ and $\Delta\sigma_n' = \Delta\sigma_n - \Delta p_{\mathrm{FP}}$, where $\Delta p_{\mathrm{FP}}$ is the overpressure on the fault plane.

## Scaling from qualitative fault stability to quantitative earthquake magnitude

With the final $CFS$ distribution along the fault, we can calculate the maximum size $S_{\max}$ of the slipping fault patch, which is defined as[68]

$$S_{\max} = \max(\ell_i)/\sin\theta, \quad (19)$$

**Table 3 | Random variables and their characteristic values for both RS-S and SS-R patterns. STD denotes standard deviation**

| Parameter | Physical meaning | Slip pattern | Minimum | Maximum | Mean | STD | Unit |
|---|---|---|---|---|---|---|---|
| $\sigma_H$ | The maximum horizontal principal stress | RS-S | 198 | 243 | 220.5 | 7.5 | MPa |
| | | SS-R | 168 | 203 | 185.5 | 5.83 | |
| $\sigma_h$ | The minimum horizontal principal stress | RS-S | 107 | 120 | 113.5 | 2.17 | MPa |
| | | SS-R | 93 | 95 | 94 | 0.33 | |
| Azimuth | Azimuth of $\sigma_H$ | RS-S | N77 | N111 | N94 | 5.67 | ° |
| | | SS-R | N74 | N100 | N87 | 4.33 | |
| $\theta$ | Fault dip | RS-S | 43 | 58 | 50.5 | 2.5 | ° |
| | | SS-R | 66 | 75 | 70.5 | 1.5 | |
| $f_{st}$ | Static friction coefficient | RS-S | 0.45 | 0.55 | 0.5 | 0.017 | - |
| | | SS-R | | | | | |
| $\alpha$ | Biot's coefficient | RS-S | 0.5 | 0.8 | 0.65 | 0.05 | - |
| | | SS-R | | | | | |

Note: the uncertainty of fault strike is covered by the azimuth of $\sigma_H$, thus, the strike is 214.5° for the RS-S pattern and 223° for the SS-R pattern.

where $l_i$ is a continuous interval along the vertical $y$ axis with $CFS > 0$. This size quantifies the portion of the fault that becomes unstable and thus provides a direct measure of the induced seismicity potential.

From the framework of fracture mechanics, the ultimate shear failure size of the fault is dominated by the fracture energy, and thus, is scaled to the seismic moment. Within this framework, a theoretical scaling relation between the seismic moment ($M_0$ in Nm) and the critical size ($2L_c$) of the nucleation zone was established by Ohnaka[71]

$$M_0 = k_{NL}(2L_c)^3, \tag{20}$$

where $k_{NL}$ is a scaling parameter. Theoretical analysis and laboratory-based observations find that $k_{NL} = 10^9$ N/m², which also aligns well with seismological data[71,72].

To bridge the result of fault stability assessment with the earthquake magnitude, we assume that the maximum size of the slipping fault patch serves as a first-order estimate of the critical nucleation dimension $2L_c$. Despite the nucleation zone extends beyond the initial slipping fault patch due to slip-weakening effects[53,54], the quasi-static shear frictional equilibrium conditions underlying both parameters enables a practical and applicable approximation to some extent. We then can quantitatively convert the seismic moment evaluated with the maximum slipping fault patch into the moment magnitude $M_w$ of earthquakes using the well-known Kanamori and Hanks relation[73,74]

$$M_w = \frac{2}{3}\log M_0 - 6.07. \tag{21}$$

Equation (21) works well only for detectable seismicity. In practice, the lower limit of the detectable seismicity size usually is about −1, which implies a 0.34 m of the critical size of the nucleation zone, i.e., the slipping fault patch size $S_{max}$. Such a value is also far below the model geometric accuracy. Therefore, for any smaller $S_{max}$ evaluated by Eq. (19), we arbitrarily assign $M_w$ as -2.

## SPA via Monte Carlo simulations

As mentioned in the Introduction, this work focuses on analyzing a prior probability of Pohang earthquake associated to the uncertainty of geomechanical properties (in-situ stress, fault orientation, and rock properties), assuming that the physical models defining the poromechanical problem are valid. To this end, we propose an SPA approach based on Monte Carlo simulations, which entails treating uncertain properties as random variables and performing a large number of realizations integrating physical models to characterize the distribution of model outputs.

We treat $\sigma_H$ and its azimuth, $\sigma_h$, $\theta$, $f_{st}$, and $\alpha$ as the random variables of this stochastic simulation (Table 3). The first four variables, covering the uncertainty of in-situ stress tensor and fault orientation, have pattern-dependent uncertainty bounds that are inferred from existing estimates with a plausibility analysis (Fig. 2). The latter two variables for rock properties share the identical uncertainty range for both slip patterns, which are estimated from experimental results. In particular, we take the laboratory test value 0.5[36] as the mean value of $f_{st}$ and apply a 10% variation around this value to cope with testing uncertainty. Such 10% uncertainty is also supported by the plausibility analysis (Fig. 2). Regarding Biot's coefficient, there is no direct tests for the Pohang case, while a value of 0.8 has been employed in some numerical simulations for Pohang earthquake[13,38]. Experiments on granite find that $\alpha$ decreases with increasing Terzaghi effective stress[75], and several cases that are close to 0.5 under high stress conditions have been reported[76,77]. Thus, we adopt 0.8 as its maximum and 0.5 as its minimum given the high in-situ stress in the source region. We assume that all these variables follow truncated normal distributions, because natural quantities often distribute approximately normally, which is supported by the central limit theorem[78,79]. With such distributions and ranges of these variables, the mean of the variables is calculated as (maximum + minimum)/2 and the standard deviation (STD) as (maximum – mean)/3, corresponding to 99.7% confidence interval based on empirical rules[79].

We then follow the standard process of Monte Carlo simulations to perform realizations starting with sampling from the defined random variables. Substituting such samples into the physical models yields the initial state of the fault, which may range from very stable to already unstable (Fig. 3). An initially very stable fault will not slip, while an initially unstable fault is already in failure. Both extreme cases are unrealistic for the Pohang EGS site, which implies invalid samples that should be discarded. We exclude these extreme cases by restricting the initial Coulomb Failure Stress ($CFS^0$) of the fault in the range of -1 to −0.01 MPa. It indicates that we are calculating probabilities conditioned to the physical models and the initial stress state (represented by $CFS^0$) being within the specified range. Therefore, our SPA method consists of the following steps to perform a large number of valid Monte Carlo realizations:

(1) Generate the sample of random variables within their truncated distributions.

(2) Evaluate $CFS^0$ of the fault and check if $CFS^0$ falls in the given range: if yes, go to the next step; if not, discard the sample and go back to the previous step.

(3) Sequentially evaluate the poroelastic stress (Eqs. (6)-(8)), the final $CFS$ (Eq. (17)), the maximum size $S_{max}$ (Eq. (19)) of the slipping fault patch, and the earthquake magnitude $M_w$ (Eq. (21)).

(4) Add the sample and evaluated results to the stack of valid realizations, update the stack mean and standard deviation of $M_w$, and plot them against the number of realizations to monitor their convergence to a plateau with increasing stack size.

(5) Repeat steps (1) to (4) to do more valid realizations until such plots reach a stationary state as the full parameter space has been explored and additional realizations would not yield any further changes.

## Data availability
All data used to evaluate the results and draw the conclusions of the paper is included in the deposited custom code, which is available at https://doi.org/10.20350/digitalCSIC/17668. Most of the data is also presented in the paper and/or the Supplementary Information. The source data regarding earthquake catalog and hydraulic stimulation is also available at https://www.science.org/doi/10.1126/science.aat6081 and https://doi.org/10.1038/s41467-020-16408-0, respectively.

## Code availability
The computer code used to create the results presented in this paper is deposited on the open repository Digital Csic https://doi.org/10.20350/digitalCSIC/17668.

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

## Acknowledgements

H.W. acknowledge the financial support of the Secretariat for Universities and Research of the Ministry of Business and Knowledge of the Government of Catalonia (AGAUR) and the European Social Fund (FI−2019). H.W. also acknowledges the Becas Santander Research Scholarship from the Technical University of Catalonia (UPC) and the Santander. V.V. acknowledges funding from the European Research Council (ERC) under the European Union's Horizon 2020 Research and Innovation Program through the Starting Grant GEoREST (www.georest.eu) under Grant agreement No. 801809 and support from the Spanish Ministry of Science, Innovation and Universities through project HydroPoreII (PID2022-137652NB-C44). IMEDEA is an accredited "Maria de Maeztu Excellence Unit" (Grant CEX2021-001198, funded by MICIU/AEI/10.13039/501100011033). V.V. and F.P. acknowledge funding from the European Union's Horizon 2020 Research and Innovation Program through the Marie Sklodowska-Curie Action ARMISTICE under grant agreement No. 882733. A.A. and P.M. acknowledge the financial support from the Swiss Federal Office of Energy within the framework of the ZoDrEx GEOTHERMICA project under contract SI/501720-01. M.S. acknowledges financial support from the "HEATSTORE" project (170153–44011), which has been subsidized through the ERANET Cofund GEOTHERMICA (Grant agreement no. 731117), from the European Commission and the Spanish Ministry of Science, Innovation and Universities (PCI2018-092933). J.C. acknowledges support of the EU through "Eastern Lights" project under Grant Agreement number 101136122.

## Author contributions

H.W., V.V., and F.P. conceived the study. H.W. collected the data, developed the calculating code, performed the simulations, and prepared the figures. V.V. supervised the research project. All authors, H.W., V.V., F.P., A.A., P.M., J.C., and M.S., contributed to the interpretation of results, writing, and editing of the paper.

## Competing interests

The authors declare no competing interests.
