## [Transparent Peer Review file · Communications Earth & Environment]

Stochastic poromechanical analysis forecasts a notable exceedance probability for the 2017 Pohang, South Korea, Mw 5.5 earthquake

Corresponding Author: Dr Haiqing Wu

Version 0:

Decision Letter:

Dear Dr Wu,

Please allow me to sincerely apologise for the long delay in sending a decision on your manuscript titled "Stochastic poromechanical analysis forecasts a significant exceedance probability of the 2017 Pohang, South Korea, Mw 5.5 earthquake". It has now been seen by 2 reviewers, and we include their comments at the end of this message. They find your work of interest, but some important points are raised. We are interested in the possibility of publishing your study in Communications Earth & Environment, but would like to consider your responses to these concerns and assess a revised manuscript before we make a final decision on publication.

In particular, please ensure that in the revised manuscript you fully justify your model assumptions and present a robust validation of your results.

We therefore invite you to revise and resubmit your manuscript, along with a point-by-point response that takes into account the points raised. Please highlight all changes in the manuscript text file.

Please submit your point-by-point responses as a separate file, distinct from your cover letter where you can add responses to the Editors' comments that you do not want to be made available to the reviewers. Word files are preferred. We recommend that any figures, tables or graphs that are included in the response to reviewers are also included in the main article or Supplementary Information.

Please use the following link to submit your revised manuscript, point-by-point response to the referees' comments (which should be in a separate document to any cover letter), a tracked-changes version of the manuscript (as a PDF file) and the completed checklist:

Link Redacted

We hope to receive your revised paper within six weeks; please let us know if you aren't able to submit it within this time so that we can discuss how best to proceed. If we don't hear from you, and the revision process takes significantly longer, we may close your file. In this event, we will still be happy to reconsider your paper at a later date, as long as nothing similar has been accepted for publication at Communications Earth & Environment or published elsewhere in the meantime.

Please do not hesitate to contact us if you have any questions or would like to discuss these revisions further. We look forward to seeing the revised manuscript and thank you for the opportunity to review your work.

Best regards,

Joe Aslin

Deputy Editor,
Communications Earth & Environment

Consulting Editor,
Communications Sustainability

<https://www.nature.com/commsenv/>
Twitter: @CommsEarth

EDITORIAL POLICIES AND FORMATTING

- Behavioural and social science
- Ecological, evolutionary & environmental sciences
- Life sciences

Furthermore, please align your manuscript with our format requirements, which are summarized on the following checklist: <https://www.nature.com/documents/commsj-phys-style-formatting-checklist-article.pdf> Communications Earth & Environment formatting checklist

and also in our style and formatting guide <https://www.nature.com/documents/commsj-phys-style-formatting-guide-accept.pdf> Communications Earth & Environment formatting guide .

*** DATA: Communications Earth & Environment endorses the principles of the Enabling FAIR data project (<http://www.copdess.org/enabling-fair-data-project/>). We ask authors to make the data that support their conclusions available in permanent, publically accessible data repositories. (Please contact the editor if you are unable to make your data available).

All Communications Earth & Environment manuscripts must include a section titled "Data Availability" at the end of the Methods section or main text (if no Methods). More information on this policy, is available at <http://www.nature.com/authors/policies/data/data-availability-statements-data-citations.pdf>.

If a community resource is unavailable, data can be submitted to generalist repositories such as <https://figshare.com/> or <http://datadryad.org/> Dryad Digital Repository. Please provide a unique identifier for the data (for example a DOI or a permanent URL) in the data availability statement, if possible. If the repository does not provide identifiers, we encourage authors to supply the search terms that will return the data. For data that have been obtained from publically available sources, please provide a URL and the specific data product name in the data availability statement. Data with a DOI should be further cited in the methods reference section.

REVIEWER COMMENTS:

Reviewer #1 (Remarks to the Author):

The authors conducted detailed analyses of stress field by considering the evolution of pore fluid pressure and stresses caused by the pressurized region to explore triggering mechanism of the 2017 Pohang, south Korea, earthquake. Since there is significant uncertainty in many model parameters such as stress field before the injection, the authors performed Monte Carlo simulations to evaluate possible earthquakes. The authors claimed that the present method may be used as physics-based risk assessment of geothermal energy projects. This is an important issue, and the authors' efforts are commendable. However, the proposed model makes many assumptions and some of them are not very reliable. Especially, the estimation method of earthquake magnitude was not fully justified. The authors argued mainly the magnitude of the mainshock. I consider that other data such as the spatial distribution of seismicity may be used to verify the model. I would like to see more careful discussion of the validity of the results.

Specific comments

Lines 234-267

The present model suggests that CFS near point P2 is increased before the mainshock. Were there foreshocks around P2? Stress distribution in Fig. 5 suggests that earthquake rupture would be arrested near point P1. The distribution of aftershocks is considered to represent the mainshock slip area. Is the aftershock distribution of the 2017 Pohang earthquake consistent with the calculated stress distribution?

Lines 281-283

The model predicted that the maximum magnitude of induced seismicity is Mw7.0. The fault length of M7 earthquake is about 30 km. Can earthquake rupture propagate to shallower than P1, where CFS is low? Is this reasonable in the present tectonic setting?

Lines 476-508

The stress field before the 2017 Pohang earthquake is important in the present analysis. Because the authors used the estimates from previous studies, brief explanation for each estimate such as the method and data is useful. Especially, the estimated areas are important, because nonuniform prestress affects the extent of rupture areas of triggered earthquakes. Uniform prestress in the study is assumed in the present model. Is this justified by observed data?

Lines 484-486

Deviatoric stress ratio defined by Equation (1) is not for the stresses on the earthquake fault plane but for the plane of the maximum shear. Is this appropriate for the present analysis?

Lines 538-542

Please explain the discretization for calculating the mean pore pressure. I think that n in Eq. (5) is different from n in Eq. (2).

Lines 546-556

When the pore pressure is a function of r , can stresses inside the inclusion be expressed by the mean pore pressure change?

Lines 593-610

The authors estimated the seismic moment by using Ohnaka's relation between the seismic moment and the nucleation zone size. I consider that a positive CFS zone is not a nucleation zone where preseismic sliding occurs. Dynamic earthquake rupture occurs when the released strain energy exceeds the fracture energy at the tip of the nucleation zone. The critical nucleation zone length is determined by the energy balance, and it is not directly related to CFS. Moreover, there are few reliable estimates of earthquake nucleation zones.

Table S3

Shear modulus of 13.8GPa seems to be too low for crystalline rock.

Reviewer #2 (Remarks to the Author):

Review of 12510 (Stochastic poromechanical analysis... by Wu et al.)

This paper reports on a Stochastic Poromechanical Analysis (SPA) approach to lessen uncertainty problems with geomechanical parameters when forecasting an exceedance probability of the 2017 Mw 5.5 Pohang earthquake triggered by EGS stimulations. Key conclusions include: a prior probability of 7–18% for the Pohang mainshock; an initial Coulomb failure stress of -0.2 to -0/1 MPa (one order of magnitude larger than that for natural earthquakes, implying that the Pohang earthquake fault was subcritically stressed); and occurrence of damaging earthquakes triggered by a small poromechanical perturbation in critically stressed faults.

The paper addresses a timely and important topic in a logical manner. It is well organized and clearly illustrated. I have no objection to its publication in *Communications Earth & Environment*, but I do have some concerns about the data handling and interpretations that should be addressed before final acceptance.

(1) The authors conclude that the prior probability of the Mw 5.5 mainshock for the RS-S pattern (6.9%) is lower than that for

the SS-R pattern (15%). However, focal mechanism solutions of the mainshock indicate a reverse-slip–dominant oblique-slip pattern (i.e., RS-S pattern; Kim et al., 2018; Ellsworth et al., 2019; Woo et al., 2019; Cho et al., 2023). How do the authors justify this apparent contradiction?

(2) The paper adopts an experimental static friction coefficient of 0.5 from direct shear tests on natural fracture surfaces of drill core retrieved at ~4200 m depth (Kwon et al., 2018). Although Kwon et al. (2018) did not identify the fracture-filling minerals, these were later found to be epidote and chlorite (An et al., 2022, *J. Geophys. Res. Solid Earth*, 127, e2021JB023136). An et al. (2022) reported a static friction coefficient of ~0.7 from shear tests on artificial epidote gouge. In contrast, drill cuttings from the ~3800 m–depth fault zone in PX-2 consist of chlorite gouge (Korean Government Commission, 2019), and Okamoto et al. (2019, *J. Geophys. Res. Solid Earth*, 124, 4545–4565) reported static friction coefficients of 0.3–0.4 and dynamic steady-state coefficients of 0.2–0.3. Given this, it would be more appropriate to adopt 0.3–0.4 as the experimental static friction coefficient.

(3) I am curious about the permeability values used for the pressurized basement rock and the causative fault. Analyses of drill cuttings and logging data (including mud losses) from PX-2 suggest that the Pohang earthquake fault consists of a fault core (gouges and cataclasites) and a damage zone with high fracture density (Korean Government Commission, 2019). Thus, the permeability structure likely comprises at least three zones: an almost impermeable fault core, a highly permeable damage zone, and host rock with in-between permeability. Based on the summary of fault and mud-loss data for depths of 3785–3840 m in PX-2 (Fig. O-3; Korean Government Commission, 2019), one can estimate the thicknesses of the fault core and damage zone. Could the authors incorporate such a more detailed permeability structure into their poromechanical modeling?

(4) Minor points

- Line 144: IS6 in Table 2 represents a regional stress state estimated from focal mechanism solutions, not a local stress state. It may therefore not be appropriate for analyses of the Pohang earthquake fault.
- Line 197: If the intermediate principal stress lies on the fault plane, it has no effect on fault stability. The phrase “a smaller effect” would be better replaced with “no or smaller effect.”
- Lines 218–219: The distances are 513 m and 128 m in Fig. 1C.
- Lines 237–239: Please explain why.
- Lines 257–258 (while...mainshock): Please elaborate on this point.
- Lines 312–314: Please explain why.

Best regards,
Jin-Han Ree

** Visit Nature Portfolio's author and referees' website at www.nature.com/authors for information about policies, services and author benefits**

Communications Earth & Environment is committed to improving transparency in authorship. As part of our efforts in this direction, we are now requesting that all authors identified as ‘corresponding author’ create and link their Open Researcher and Contributor Identifier (ORCID) with their account on the Manuscript Tracking System prior to acceptance. ORCID helps the scientific community achieve unambiguous attribution of all scholarly contributions. You can create and link your ORCID from the home page of the Manuscript Tracking System by clicking on ‘Modify my Springer Nature account’ and following the instructions in the link below. Please also inform all co-authors that they can add their ORCIDs to their accounts and that they must do so prior to acceptance.

If you experience problems in linking your ORCID, please contact the Platform Support Helpdesk.

Version 1:

Decision Letter:

Dear Dr Wu,

Your manuscript titled "Stochastic poromechanical analysis forecasts a significant exceedance probability of the 2017 Pohang, South Korea, Mw 5.5 earthquake" has now been seen by our reviewers, whose comments appear below. In light of

their advice we are delighted to say that we are happy, in principle, to publish a suitably revised version in Communications Earth & Environment.

We therefore invite you to revise your paper one last time to address the remaining concerns of our reviewers. At the same time we ask that you edit your manuscript to comply with our format requirements and to maximise the accessibility and therefore the impact of your work.

EDITORIAL REQUESTS:

*****Please take care to match our formatting and policy requirements. We will check revised manuscript and return manuscripts that do not comply. Such requests will lead to delays. *****

SUBMISSION INFORMATION:

OPEN ACCESS:

Communications Earth & Environment is a fully open access journal. Articles are made freely accessible on publication. For further information about article processing charges, open access funding, and advice and support from Nature Portfolio, please visit <https://www.nature.com/commsenv/open-access>

Link Redacted

We hope to hear from you within four weeks; please let us know if you need more time.

Best regards,

Joe Aslin

Deputy Editor,
Communications Earth & Environment

Consulting Editor,
Communications Sustainability

<https://www.nature.com/commsenv/>
Twitter: @CommsEarth

REVIEWERS' COMMENTS:

Reviewer #1 (Remarks to the Author):

The authors carefully responded to all the comments, and the manuscript was significantly improved. I think that the revised version is acceptable. Although I don't necessarily agree with relating the maximum size of the slipping fault patch to the critical nucleation dimension, I think the revised version is an improvement because it is worded more carefully.

Small comments
Line 402

'SS-R patter' may be 'SS-R pattern.'

Equation (5)

In the denominator of the right-hand side, 'n' may be '1.'

Reviewer #2 (Remarks to the Author):

Review of COMSENV-25-2365A (Stochastic poromechanical analysis... by Wu et al.)

I have reviewed the revised manuscript, "Stochastic poromechanical analysis forecasts a significant exceedance probability of the 2017 Pohang, South Korea, Mw 5.5 earthquake", by Wu et al. The authors have addressed all of my comments, and I believe the manuscript is now ready for publication.

Jin-Han Ree

** Visit Nature Portfolio's author and referees' website at www.nature.com/authors for information about policies, services and author benefits**

Response to Reviewers' Comments on the Manuscript "COMMSENV-25-2365"

We discuss below the comments made by the reviewers and how we have responded to them point-by-point. To facilitate reading, we have pasted the original comments in *italics* and our responses in blue font. All the line numbers cited in our responses refer to the revised manuscript with tracked changes (changes are marked in red).

Reviewers' comments:

Reviewer #1 (Remarks to the Author):

The authors conducted detailed analyses of stress field by considering the evolution of pore fluid pressure and stresses caused by the pressurized region to explore triggering mechanism of the 2017 Pohang, south Korea, earthquake. Since there is significant uncertainty in many model parameters such as stress field before the injection, the authors performed Monte Carlo simulations to evaluate possible earthquakes. The authors claimed that the present method may be used as physics-based risk assessment of geothermal energy projects. This is an important issue, and the authors' efforts are commendable. However, the proposed model makes many assumptions and some of them are not very reliable. Especially, the estimation method of earthquake magnitude was not fully justified. The authors argued mainly the magnitude of the mainshock. I consider that other data such as the spatial distribution of seismicity may be used to verify the model. I would like to see more careful discussion of the validity of the results.

Response: We sincerely thank the reviewer for assessing in detail this manuscript and his/her positive assessment of our work. We agree that the reviewer's main concern is as important as inherently complex. Addressing such complexity requires some simplifying assumptions, yet we emphasize that our approach is grounded in well-established physical principles: Hooke's Law and Coulomb's failure criterion, Darcy's law and poroelasticity. The main novelty of our work lies not in redefining these physics, but in embedding a traditionally deterministic analysis within a stochastic framework. Given the substantial uncertainty associated with geomechanical properties and fault geometry, we believe this probabilistic characterization is not only appropriate but necessary for realistic seismic hazard assessment.

Regarding the reliability of the method for estimating earthquake magnitude, we provide further clarification and justification in our response to the specific comment on lines 593-610 (see below). We also greatly appreciate the reviewer's suggestion regarding the verification of our model and results, which is valuable for improving the quality of this manuscript. Details for such verification, based on the tectonic setting and the observed seismicity, are provided in our response to the next comment.

Specific comments

Lines 234-267

The present model suggests that CFS near point P2 is increased before the mainshock. Were there foreshocks around P2?

Stress distribution in Fig. 5 suggests that earthquake rupture would be arrested near point P1. The distribution of aftershocks is considered to represent the mainshock slip area. Is the aftershock distribution of the 2017 Pohang earthquake consistent with the calculated stress distribution?

Response: We thank the reviewer for this insightful comment that would be very helpful to further verify our models and results. The reviewer's interpretation is correct, and our answer is yes for both questions raised. The hypocenters of the six foreshocks that occurred about nine hours before the mainshock are all clustered in close proximity to the cross-point P2 (4.65 km, Kim et al., 2018; Ellsworth et al., 2019; Lee, 2019). Moreover, most hypocentral depths of the 210 aftershocks recorded during the first three hours following the mainshock fall within the 4-6 km range, and the mainshock depth itself is estimated at roughly 4.5 km (Kim et al., 2018; Ellsworth et al., 2019; Lee, 2019). This spatial feature is highly consistent with the distribution of Coulomb Failure Stress changes presented in Figure 5C.

To strengthen the verification of our fault reactivation model, we have now incorporated the hypocenters of the main foreshocks, the mainshock, and these early aftershocks into Figure 5C and added an explanatory discussion in the revised manuscript (lines 260-264 and 388-390). These updates highlight that the less stable fault patch predicted by our physics-based approach accurately reproduces the observed seismicity patterns.

Lines 281-283

The model predicted that the maximum magnitude of induced seismicity is Mw7.0. The fault length of M7 earthquake is about 30 km. Can earthquake rupture propagate to shallower than P1, where CFS is low? Is this reasonable in the present tectonic setting?

Response: The fault length of 30 km required to trigger a Mw 7 earthquake would necessarily extend in the out-of-plane direction (i.e., along fault strike), orthogonal to the cross section illustrated in Figure 1C. Observations of the induced seismicity at Pohang indeed show a seismic cloud stretching 3 to 18 km in this out-of-plane direction (Grigoli et al., 2018; Kim et al., 2018; Ellsworth et al., 2019; Lee, 2019), depending on the temporal window considered. Specifically, the cloud extends roughly 3 km during the first three hours of aftershocks (Kim et al., 2018), 10 km after one day (Ellsworth et al., 2019; Lee, 2019), and up to 18 km after 15 days of aftershocks (Grigoli et al., 2018), spatially aligning with the rupture of the Mw 5.5 mainshock. These observations suggest that, given the tectonic architecture of the Pohang basin (Grigoli et al., 2018; Lee, 2019), a larger rupture could plausibly propagate further along strike in principle for an event of magnitude Mw 7. However, our stochastic analysis (Figure 6) indicates that the probability of inducing such a Mw 7 event is essentially negligible, and even the probability of reaching $Mw \geq 5.5$ is only about 7%-15%.

In contrast, the in-plane poromechanical perturbation is tightly concentrated around the P1-P2 segment of the fault (Figures 1 and 5), in which the Coulomb Failure Stress

changes (ΔCFS) are near zero or even negative above P1 (Figure 5C). As a consequence, the rupture propagation would not extend to the shallow formations above P1. Instead, it can propagate to the deep layers below the cross-point P2. This is consistent with both the tectonic setting and the observed foreshocks (Figure 5C). We note, however, that the rupture may propagate to depths shallower above P1 when considering dynamic rupture processes caused by the slip-weakening mechanism (Rice and Ruina, 1983; Rice et al., 2005; Garagash and Germanovich, 2012), which are not included in the present modeling.

Lines 476-508

The stress field before the 2017 Pohang earthquake is important in the present analysis. Because the authors used the estimates from previous studies, brief explanation for each estimate such as the method and data is useful. Especially, the estimated areas are important, because nonuniform prestress affects the extent of rupture areas of triggered earthquakes. Uniform prestress in the study is assumed in the present model. Is this justified by observed data?

Response: We thank again the reviewer for this suggestion. IS1 and IS3 were derived primarily from observed data obtained in PX-2 (Ellsworth et al., 2019; Lee, 2019), which we believe can be considered fully justified; IS2, and IS4-IS6 were deduced from focal mechanism solutions of prior earthquakes, with IS5 showing good agreement with the kinematic indicators of Quaternary faults within the Pohang basin (Westaway & Burnside, 2019), and the other stress estimates aligning with the InSAR and moment tensor analyses of the 2017 Pohang earthquake in terms of faulting regimes (Grigoli et al., 2018). We have added a brief explanation summarizing the existing stress estimates including the methods used for their derivation and the underlying data, at lines 539-549.

Indeed, we also fully acknowledge that the stress field is non-uniform in reality, leading to initial stress heterogeneity that can affect the extent of rupture areas. However, all the proposed stress estimates necessarily assume uniformity because the available data does not allow reliably reconstructing a non-uniform stress field at the Pohang site (Lee, 2019). If such detailed prestress becomes available in the future (which we find difficult, even hardly possible), our analysis could be readily updated because all evaluations are analytical and the prestress affects only the distribution of the initial Coulomb Failure Stress along the fault.

Lines 484-486

Deviatoric stress ratio defined by Equation (1) is not for the stresses on the earthquake fault plane but for the plane of the maximum shear. Is this appropriate for the present analysis?

Response: Exactly, and thus, we refer to it as the maximum deviatoric stress ratio (DSR_{\max}) to avoid misunderstandings. As explained in the manuscript, we define DSR_{\max} as a first-order assessment of the proposed stress tensors. The underlying logic is straightforward: a stress tensor estimate can be ruled out for the Pohang site if the plane of the maximum shear is not critically stressed compared to the reference value of the static friction,

because no other orientation will be more critically stressed than the maximum shear. According to Equation (1), we exclude three stress estimates from Table 2, prior to the detailed plausibility analysis shown in Figure 2. This step is important to avoid unnecessary complexity, as including those three unlikely cases would require three additional subplots in Figure 2, none of them providing meaningful insight. Nonetheless, following the reviewer's comment, we have made a slight modification at lines 560-562 to articulate this rationale more explicitly.

Lines 538-542

Please explain the discretization for calculating the mean pore pressure. I think that n in Eq. (5) is different from n in Eq. (2).

Response: We thank the reviewer for pointing out this confusion. The reviewer is right, the symbol n has different meanings in Equations (5) and (2). We have better explained these meanings at lines 581 and 512, and also changed the second n to l to avoid confusion. To calculate the mean pore pressure change, we vertically and equally divided the entire pressurized region (which has a uniform thickness h) into l sub-regions, with each of them being characterized by a geometric width dr and a pressure-weighted height of $\Delta p_i/h$, where Δp_i is the overpressure of the i -th sub-region. This means that the area of every sub-region equals $\Delta p_i h dr$. Then, we calculate the mean pore pressure change averaged by the pressure-weighted area, which provides a more accurate result than that averaged using only the geometric area. We have improved the explanation of the discretization at lines 612-614.

Lines 546-556

When the pore pressure is a function of r , can stresses inside the inclusion be expressed by the mean pore pressure change?

Response: This is a good question. For accurate modeling, the poroelastic stress should be ideally evaluated with non-uniform overpressures rather than a mean value, which can also be achieved with the analytical solution applied in this work through the superposition principle. However, we did not implement that approach in the current work because the errors associated with using the mean pore pressure change are relatively small (De Simone et al., 2019), especially after long times of diffusion. Furthermore, the computational cost is significantly lower than evaluating stress changes with non-uniform overpressures, as in the current approach. Thus, we adopt the mean pressure change as an approximation to estimate the poroelastic stress changes to speed up the stochastic poromechanical simulations, which allows us to properly explore the parametric space with a large number of Monte Carlo realizations in a reasonable amount of time. Nonetheless, even using a non-uniform overpressure field would be an approximation to the actual perturbation because, e.g., heterogeneity of the permeability field (amongst others) is not considered (see Alcolea et al., 2024).

Lines 593-610

The authors estimated the seismic moment by using Ohnaka's relation between the seismic moment and the nucleation zone size. I consider that a positive CFS zone is not a nucleation zone where preseismic sliding occurs. Dynamic earthquake rupture occurs when the released strain energy exceeds the fracture energy at the tip of the nucleation zone. The critical nucleation zone length is determined by the energy balance, and it is not directly related to CFS. Moreover, there are few reliable estimates of earthquake nucleation zones.

Response: The reviewer correctly recognizes how we link the fault stability analysis to the earthquake magnitude estimate, and also the physics of dynamic rupture. We agree that dynamic rupture and the critical nucleation zone are related to the energy balance with slip-weakening mechanism (Rice and Ruina, 1983; Rice et al., 2005; Garagash and Germanovich, 2012; Ohnaka, 2000; 2013), which differ from the fault instability and the positive *CFS* zone (i.e., the slipping fault patch), respectively. However, given the simplifications made in our model, and in the absence of a more rigorous method, we take the maximum slipping fault patch as a proxy for the critical nucleation zone. We find this approximation reasonable (to some extent) because both are defined under quasi-static shear frictional equilibrium conditions, and dynamic rupture is extended from the slipping fault patch by acknowledging slip-weakening effects (Garagash and Germanovich, 2012; Azad et al., 2017; Wu et al., 2024a; 2024b). Accordingly, we have already discussed this limitation of the method and acknowledged potential extensions and improvements for future studies at lines 471-472. In response to the reviewer's comment, we have refined the statements at lines 675-686 to further clarify this assumption.

As for reliable estimates of the nucleation zone, we concur that they are limited, but still valuable. For instance, Guglielmi et al. (2015) carried out in-situ experiments at the Low-Noise Underground Laboratory in southeastern France to investigate fluid injection-induced aseismic slip, directly and clearly capturing both the nucleation zone and the nucleation process. These observations were later fit with a circular-crack interfacial slip model governed by linear slip-weakening friction (Bhattacharya and Viesca, 2019). Laboratory results of Acosta et al. (2019) revealed a similar scaling relationship to that of Ohnaka (2000), and also resolved the nucleation zone. In the context of natural earthquakes, tens of cases summarized in Ellsworth and Beroza (1995) are widely regarded as reliable. More details regarding this discussion can also be found in Wu et al. (2024a; 2024b).

Table S3

Shear modulus of 13.8 GPa seems to be too low for crystalline rock.

Response: Certainly. Typical values for crystalline rocks are about 30 GPa, but the value utilized here comes directly from laboratory testing of Pohang drilling core samples, as reported in Kwon et al. (2019), which has also been used in several other studies. Thus, since site-specific data is available, we adopt this value.

Reviewer #2 (Remarks to the Author):

Review of 12510 (Stochastic poromechanical analysis... by Wu et al.)

This paper reports on a Stochastic Poromechanical Analysis (SPA) approach to lessen uncertainty problems with geomechanical parameters when forecasting an exceedance probability of the 2017 Mw 5.5 Pohang earthquake triggered by EGS stimulations. Key conclusions include: a prior probability of 7–18% for the Pohang mainshock; an initial Coulomb failure stress of -0.2 to -0/1 MPa (one order of magnitude larger than that for natural earthquakes, implying that the Pohang earthquake fault was subcritically stressed); and occurrence of damaging earthquakes triggered by a small poromechanical perturbation in critically stressed faults.

The paper addresses a timely and important topic in a logical manner. It is well organized and clearly illustrated. I have no objection to its publication in Communications Earth & Environment, but I do have some concerns about the data handling and interpretations that should be addressed before final acceptance.

Response: We thank the reviewer for these positive words and for the detailed review as demonstrated by the highlighted key conclusions attained from this work. Regarding reviewer's concerns about the data handling and interpretations, we have provided point-by-point responses to the detailed comments below and incorporated the corresponding reviews in the new version of the manuscript.

(1) The authors conclude that the prior probability of the Mw 5.5 mainshock for the RS-S pattern (6.9%) is lower than that for the SS-R pattern (15%). However, focal mechanism solutions of the mainshock indicate a reverse-slip–dominant oblique-slip pattern (i.e., RS-S pattern; Kim et al., 2018; Ellsworth et al., 2019; Woo et al., 2019; Cho et al., 2023). How do the authors justify this apparent contradiction?

Response: The oblique-slip pattern that better correlates with the mainshock depends on the stress faulting regime and fault orientation. Existing data and analyses indicate that both RS-S and SS-R patterns are possible (Tables 1 and 2, and references therein). We agree that the focal mechanism solutions support the RS-S more than the SS-R pattern (Kim et al., 2018; Ellsworth et al., 2019; Woo et al., 2019; Cho et al., 2023). However, the proposed stochastic poromechanical analysis should be performed before operations start, so prior to the occurrence of an earthquake. Moreover, focal mechanism solutions are not uncertainty-free and, correspondingly, we cannot rule out the SS-R pattern in our analysis.

Regarding our stochastic results, it is true that the exceedance probability of the mainshock for the RS-S pattern is lower than for the SS-R pattern, but it does not indicate a strong preference for the SS-R pattern because both slip patterns yield outcomes consistent with the frequency-magnitude relationship of recorded seismicity and both with a reasonable b -value (Shapiro et al., 2021). Therefore, the stochastic poromechanical analysis alone cannot discriminate which slip pattern is more likely at Pohang, as we already noted in the first paragraph (at lines 404-405) of the Discussion Section, where we stated that additional site-specific constraints are required. Thus, there is no

contradiction. To concur with the reviewer, we have added a clarifying sentence at lines 405-406.

(2) The paper adopts an experimental static friction coefficient of 0.5 from direct shear tests on natural fracture surfaces of drill core retrieved at ~4200 m depth (Kwon et al., 2018). Although Kwon et al. (2018) did not identify the fracture-filling minerals, these were later found to be epidote and chlorite (An et al., 2022, J. Geophys. Res. Solid Earth, 127, e2021JB023136). An et al. (2022) reported a static friction coefficient of ~0.7 from shear tests on artificial epidote gouge. In contrast, drill cuttings from the ~3800 m–depth fault zone in PX-2 consist of chlorite gouge (Korean Government Commission, 2019), and Okamoto et al. (2019, J. Geophys. Res. Solid Earth, 124, 4545–4565) reported static friction coefficients of 0.3–0.4 and dynamic steady-state coefficients of 0.2–0.3. Given this, it would be more appropriate to adopt 0.3–0.4 as the experimental static friction coefficient.

Response: We thank the reviewer for the additional information regarding the friction coefficient. This coefficient is indeed challenging to determine in deep geothermal engineering. In fact, it is usually highly heterogeneous at the local scale, as demonstrated in the work by Dahrabou et al. (2021) using breakout geometry along the borehole BS-1 in Basel. For the Pohang case, the most relevant measurement is the one presented in Kwon et al. (2019, which was published online in 2018). We concur that Kwon et al. (2019) did not identify the fracture-filling minerals, i.e., the fault gouge, but the drill cores clearly show the presence of chlorite, as shown in Fig. 5 of Kwon et al. (2019) and Fig. 1 of An et al. (2022). It is true that Okamoto et al. (2019) reported relatively low values of the static friction coefficient, but those experiments were performed with pure chlorite samples originating from Malacacheta, Minas Gerais, Brazil, which may do not represent that in Pohang. Additionally, An et al. (2025) showed that the friction coefficient decreases with increasing chlorite content. However, the fault gouge at Pohang contains only about 10-15% of chlorite (Lee, 2019), suggesting that the static friction coefficient of the ruptured fault associated with the Pohang earthquake may be not as low as indicated by Okamoto et al. (2019). Therefore, we believe that taking the measurement of Kwon et al. (2019) as a base value of the static friction for Pohang is a better option. We have added this clarification in the revised manuscript at lines 556-559. Nonetheless, our stochastic analysis explores the friction coefficient in the range 0.45 to 0.55. The lower bound approximately aligns with the reviewer's suggestion.

(3) I am curious about the permeability values used for the pressurized basement rock and the causative fault. Analyses of drill cuttings and logging data (including mud losses) from PX-2 suggest that the Pohang earthquake fault consists of a fault core (gouges and cataclasites) and a damage zone with high fracture density (Korean Government Commission, 2019). Thus, the permeability structure likely comprises at least three zones: an almost impermeable fault core, a highly permeable damage zone, and host rock with in-between permeability. Based on the summary of fault and mud-loss data for depths of 3785–3840 m in PX-2 (Fig. O-3; Korean Government Commission, 2019), one can

estimate the thicknesses of the fault core and damage zone. Could the authors incorporate such a more detailed permeability structure into their poromechanical modeling?

Response: We thank the reviewer for this insightful comment. The three zones of fault structure in terms of permeability are indeed the actual situation at Pohang. Unfortunately, such detailed fault structure is typically implemented only in numerical models, as it poses a challenge for analytical solutions. In this work, we apply analytical solutions to represent the hydro-mechanical processes during and after injection, which enables us to conduct a computationally efficient stochastic poromechanical analysis with a large number of realizations to address uncertainties of geomechanical properties and fault orientation. Given this framework, we currently cannot exactly incorporate the detailed fault structure into our modeling. Nonetheless, we have considered a constant permeability $5 \times 10^{-18} \text{ m}^2$ for the pressurized basement rock (Table S3), treated the fault core as impermeable, and acknowledged the impact of the highly permeable fault damage zone on pore pressure diffusion. In particular, we assume that the injection-induced overpressure can dissipate through such damage zone along the longitudinal direction, and thus, no more additional overpressure generates in the injection side of the fault and no pressure perturbation occurs in the other side. We have better clarified this fault structure and the related assumption at lines 225-230 and 596-604 of the revised manuscript. A numerical study including a heterogeneous permeability field and the fluid-induced permeability changes, derived from PX-2 observations, can be found in Alcolea et al. (2024).

(4) Minor points

- Line 144: IS6 in Table 2 represents a regional stress state estimated from focal mechanism solutions, not a local stress state. It may therefore not be appropriate for analyses of the Pohang earthquake fault.

Response: We thank the reviewer for highlighting this detailed information regarding the stress state estimate IS6. Indeed, IS6 was estimated from prior earthquakes over the entire Korean Peninsula rather than specifically those from the Pohang basin (Soh et al., 2018). We did not exclude IS6 from the first-order assessment using Equation (1) because it yields a high value of the maximum deviatoric stress ratio (DSR_{\max}). However, IS6 was ruled out after the subsequent plausibility analysis (Figure 2), which thus provides an additional and independent verification of the goodness of our analysis. We have incorporated this clarification into the manuscript at lines 162-163.

- Line 197: If the intermediate principal stress lies on the fault plane, it has no effect on fault stability. The phrase “a smaller effect” would be better replaced with “no or smaller effect.”

Response: We agree that the intermediate principal stress has no effect on the fault when it is parallel to the fault strike. However, it is also true that the maximum or the minimum principal stress has also no effect on the fault when it is parallel to the fault strike. Thus, the impact of principal stresses (any) depends on the stress azimuth and the fault strike. Here, our intention is to convey that the intermediate principal stress generally has a smaller influence on the potential shear failure of rocks than the maximum or minimum

principal stresses under typical conditions, which helps to support and explain our results. Accordingly, we have replaced the phrase “fault stability” with “potential shear failure of rocks” at line 207 to make this statement more general and precise.

- Lines 218–219: *The distances are 513 m and 128 m in Fig. 1C.*

Response: Exactly. Those distances correspond to the minimum and maximum dip angles, respectively, as shown in Fig. 1C. Here, we are analyzing the base scenario with mean values of in-situ stress and fault orientation, which results in a distance of 394 m for the RS-S pattern and 169 m for the SS-R pattern, depending on the dip angle. We recognize that citing Fig. 1C here may cause confusion, so we have rephrased the sentence at lines 236-237 to clarify this point.

- Lines 237–239: *Please explain why.*

Response: We can interpret the stabilizing and destabilizing effects of poroelastic stress through the definition of Coulomb Failure Stress changes (ΔCFS , Equation (18)). According to this definition, negative variations in shear stress reduce the driving force of shear failure, whereas positive variations in normal stress increase the slip resistance, thereby stabilizing the fault (precisely the situation at the cross-point P1). Conversely, the aforementioned stress variations reverse at P2, leading to fault destabilization there. We have added this explanation at line 257.

- Lines 257–258 (*while...mainshock*): *Please elaborate on this point.*

Response: Here, the final state of the fault is represented by the *CFS* value at the instant of the mainshock, calculated as the sum of the initial *CFS* and ΔCFS , which is less than -1 MPa. We have clarified this at the beginning of this paragraph. Such a negative value of final *CFS* indicates that the fault remains stable at the time of the mainshock using the base scenario with mean values of geomechanical properties and fault orientation, which is not realistic. This contradiction highlights that the mean values of geomechanical properties and fault orientation do not represent the actual situation, which motivates our stochastic analysis. We have revised this explanation at lines 275-282 to improve clarity.

- Lines 312–314: *Please explain why.*

Response: The observed difference in the maximum stability changes between the two slip patterns is primarily governed by the fault dips in our simulations. As illustrated in Figure 5, a higher dip angle (70.5° in the SS-R pattern) results in a smaller variation the absolute value of shear stress ($|\tau^f| - |\tau^0|$) after stress transformation compared to a lower dip angle (50.5° in the RS-S pattern). We have clarified this at lines 350-352.

References

Acosta, M., Passelègue, F. X., Schubnel, A., Madariaga, R., & Violay, M., 2019. Can precursory moment release scale with earthquake magnitude? A view from the laboratory. *Geophysical Research Letters*, 46(22), 12927-12937.

- Alcolea, A., Meier, P., Vilarrasa, V., Olivella, S. and Carrera, J., 2024. Hydromechanical modeling of the hydraulic stimulations in borehole PX2 (Pohang, South Korea). *Geothermics* 120, 103009.
- An, M., Elsworth, D., Zhang, W., Wang, T., Huang, R., Huang, L. and Zhang, F., 2025. Role of Chlorite on the Friction and Stability of Granite Faults and Implications for Seismicity in Deep Geothermal Reservoirs. *Rock Mechanics and Rock Engineering*, pp.1-21.
- An, M., Zhang, F., Min, K.B., Elsworth, D., He, C. and Zhao, L., 2022. Frictional stability of metamorphic epidote in granitoid faults under hydrothermal conditions and implications for injection-induced seismicity. *Journal of Geophysical Research: Solid Earth*, 127(3), p.e2021JB023136.
- Azad, M., Garagash, D. I. & Satish, M., 2017. Nucleation of dynamic slip on a hydraulically fractured fault. *J. Geophys. Res. Solid Earth* 122, 2812–2830
- Bhattacharya, P., and Viesca, R. C., 2019. Fluid-induced aseismic fault slip outpaces pore-fluid migration. *Science*, 364(6439), 464-468.
- Cho, E., Woo, J.U., Rhie, J., Kang, T.S. and Baag, S.Y., 2023. Rupture Process of the 2017 M w 5.5 Pohang, South Korea, Earthquake via an Empirical Green's Function Method. *Bulletin of the Seismological Society of America*, 113(2), pp.592-603.
- Dahrabou A., Valley B., Meier P., Brunner Ph., Alcolea A. (2021) A systematic methodology to calibrate wellbore failure models, estimate the in-situ stress tensor and evaluate wellbore cross-sectional geometry. *International Journal of Rock Mechanics & Mining Sciences*, 149 (2022) 104935.
- De Simone, S., Jackson, S.J. and Krevor, S., 2019. The error in using superposition to estimate pressure during multisite subsurface CO2 storage. *Geophysical Research Letters*, 46(12), pp.6525-6533.
- Ellsworth, W.L. and Beroza, G.C., 1995. Seismic evidence for an earthquake nucleation phase. *Science*, 268(5212), pp.851-855.
- Ellsworth, W.L., Giardini, D., Townend, J., Ge, S. and Shimamoto, T., 2019. Triggering of the Pohang, Korea, earthquake (M w 5.5) by enhanced geothermal system stimulation. *Seismological Research Letters*, 90(5), pp.1844-1858.
- Garagash, D. I., and Germanovich, L. N., 2012. Nucleation and arrest of dynamic slip on a pressurized fault. *Journal of Geophysical Research: Solid Earth*, 117(B10).
- Grigoli, F., Cesca, S., Rinaldi, A.P., Manconi, A., Lopez-Comino, J.A., Clinton, J.F., Westaway, R., Cauzzi, C., Dahm, T. and Wiemer, S., 2018. The November 2017 M w 5.5 Pohang earthquake: A possible case of induced seismicity in South Korea. *Science*, 360(6392), pp.1003-1006.
- Guglielmi, Y., Cappa, F., Avouac, J.P., Henry, P. and Elsworth, D., 2015. Seismicity triggered by fluid injection–induced aseismic slip. *Science*, 348(6240), pp.1224-1226.
- Kim, K.H., Ree, J.H., Kim, Y., Kim, S., Kang, S.Y. and Seo, W., 2018. Assessing whether the 2017 M w 5.4 Pohang earthquake in South Korea was an induced event. *science*, 360(6392), pp.1007-1009.
- Kwon, S., Xie, L., Park, S., Kim, K.I., Min, K.B., Kim, K.Y., Zhuang, L., Choi, J., Kim, H. and Lee, T.J., 2019. Characterization of 4.2-km-deep fractured granodiorite cores from Pohang Geothermal Reservoir, Korea. *Rock Mechanics and Rock Engineering*, 52(3), pp.771-782.
- Lee, K. K., 2019. Final Report of the Korean Government Commission on Relations between the 2017 Pohang Earthquake and EGS Project. <https://doi.org/10.22719/KETEP-2019043001>.

- Ohnaka, M. A Physical Scaling Relation Between the Size of an Earthquake and its Nucleation Zone Size. *Pure Appl. Geophys.* 157, 2259–2282 (2000).
- Ohnaka, M. *The Physics of Rock Failure and Earthquakes.* (Cambridge University Press, 2013).
- Okamoto, A.S., Verberne, B.A., Niemeijer, A.R., Takahashi, M., Shimizu, I., Ueda, T. and Spiers, C.J., 2019. Frictional properties of simulated chlorite gouge at hydrothermal conditions: Implications for subduction megathrusts. *Journal of Geophysical Research: Solid Earth*, 124(5), pp.4545-4565.
- Rice, J.R., Ruina, A.L., 1983. Stability of Steady Frictional Slipping. *J. Appl. Mech.* 50, 343–349. <https://doi.org/10.1115/1.3167042>
- Rice, J.R., Sammis, C.G. and Parsons, R., 2005. Off-fault secondary failure induced by a dynamic slip pulse. *Bulletin of the Seismological Society of America*, 95(1), pp.109-134.
- Soh, I., Chang, C., Lee, J., Hong, T.-K. & Park, E.-S., 2018. Tectonic stress orientations and magnitudes, and friction of faults, deduced from earthquake focal mechanism inversions over the Korean Peninsula. *Geophys. J. Int.* 213, 1360–1373.
- Westaway, R. & Burnside, N. M., 2019. Fault “Corrosion” by Fluid Injection: A Potential Cause of the November 2017 Mw 5.5 Korean Earthquake. *Geofluids* 2019, e1280721
- Woo, J.U., Kim, M., Sheen, D.H., Kang, T.S., Rhie, J., Grigoli, F., Ellsworth, W.L. and Giardini, D., 2019. An in-depth seismological analysis revealing a causal link between the 2017 MW 5.5 Pohang earthquake and EGS project. *Journal of Geophysical Research: Solid Earth*, 124(12), pp.13060-13078.
- Wu, H., Parisio, F., and Vilarrasa, V., 2024a. Poroelastic effects on the nucleation process of dynamic fault rupture during fluid injection. *GEoREST Workshop on Induced Seismicity*, Palma, Spain, 11-13 March 2024.
- Wu, H., Parisio, F., and Vilarrasa, V., 2024b. Poroelastic effects on the nucleation process of dynamic fault rupture during fluid injection. Under review in *Journal of Geophysical Research: Solid Earth*.
- Shapiro, S.A., Kim, K.-H., Ree, J.-H., 2021. Magnitude and nucleation time of the 2017 Pohang Earthquake point to its predictable artificial triggering. *Nat. Commun.* 12, 6397. <https://doi.org/10.1038/s41467-021-26679-w>

Response to Reviewers' Comments on the Manuscript "COMMSENV-25-2365A"

We discuss below the comments made by the reviewers and how we have responded to them point-by-point. To facilitate reading, we have pasted the original comments in *italics* and our responses in **blue** font.

Reviewers' comments:

Reviewer #1 (Remarks to the Author):

The authors carefully responded to all the comments, and the manuscript was significantly improved. I think that the revised version is acceptable. Although I don't necessarily agree with relating the maximum size of the slipping fault patch to the critical nucleation dimension, I think the revised version is an improvement because it is worded more carefully.

Response: We sincerely thank the reviewer for assessing in detail our revised manuscript and our responses to the reviewer's comments. We appreciate the reviewer's recognition to our work.

Small comments

Line 402: 'SS-R patter' may be 'SS-R pattern.'

Equation (5): In the denominator of the right-hand side, 'n' may be 'l.'

Response: We thank the reviewer for these careful notes. We have corrected them.

Reviewer #2 (Remarks to the Author):

Review of COMMSENV-25-2365A (Stochastic poromechanical analysis... by Wu et al.)

I have reviewed the revised manuscript, "Stochastic poromechanical analysis forecasts a significant exceedance probability of the 2017 Pohang, South Korea, Mw 5.5 earthquake", by Wu et al. The authors have addressed all of my comments, and I believe the manuscript is now ready for publication.

Response: We sincerely thank the reviewer for these positive words and for his efforts and time in reviewing this manuscript. It is our honor to get the recognition of the reviewer.